# Dynamics-Informed Weight Diffusion for Generalizable Prediction of Complex Systems

## Abstract

Data-driven methods offer an effective equation-free solution for predicting physical dynamics. However, predictive models often fail to generalize to unseen environments due to varying dynamic behaviors. In this work, we introduce DynaDiff, a novel generative meta-learning framework to enable efficient, test-time adaptation. Instead of tuning a pre-trained model or context, DynaDiff directly generates a complete, high-performance expert model from scratch, conditioned on a short observation sequence from a new target environment. Specifically, we first finetune a base model on various source environments to efficiently construct a model zoo of expert predictors. Subsequently, we leverage a weight graph representation and train a conditional diffusion model to learn the underlying distribution of expert weights, capable of generating new models from a given dynamic behavior. To effectively capture the dynamic context from the observation sequence, we design a dynamics-informed prompter that explicitly models the relationship between the system's state and its temporal evolution, providing a highly informative prompt for the generative process. Extensive experiments demonstrate that our method can generate expert models with strong generalization for new environments, conditioned on limited observations. Code is available at https://anonymous.4open.science/r/DynaDiff-8B1C/README.md.

## 1 Introduction

Data-driven approaches have emerged as a powerful, equation-free paradigm for predicting physical dynamics (Wang et al., 2023; Ding et al., 2024), achieving considerable success across a diverse range of disciplines, including molecular dynamics (Mardt et al., 2018), fluid mechanics (Shu et al., 2023), and climate science (Bi et al., 2023). In these systems, dynamical systems governed by the same underlying equations can exhibit vastly different evolutionary behaviors under varying environmental conditions $e$, which can be formally expressed as $\frac{dx}{dt} = f(x, t, e)$. For instance, fluid flows, described by the Navier-Stokes equations, can exhibit different vortex structures under various Reynolds number or external driving forces. Consequently, a predictive model $f_{\theta, e_a}$, trained on observed trajectories of a specific environmental condition $e_a$ struggles to generalize to unseen environmental conditions $e_b$. Therefore, modeling the generalizable function $f$ beyond the specific environment remains a critical problem for scientific machine learning (Subramanian et al., 2023; Goswami et al., 2022).

Significant efforts have been undertaken to enable cross-environment prediction. Meta-learning approaches facilitate adaptation to unseen environments by simultaneously learning both environment-shared weights and environment-specific contexts (Kirchmeyer et al., 2022; Wang et al., 2022; Blanke & Lelarge, 2024). When applied to a new environment, the environment-specific contexts are tuned on new data to compose or modulate a tailored predictive model. Another strategy is to train environment-unified foundation models through well-designed architectures and large-scale parameterization (Herde et al., 2024; Hao et al., 2024; McCabe et al., 2024; Yang et al., 2023; Chen et al., 2024b). These models, pretrained on massive datasets, can be partially refined by finetuning on data specific to a target environment. However, from a model weight perspective, the essence of these methods only permit adaptation within a small, expert-specified subset of weights. This approach restricts the model's ability to represent the true, complex manifold of expert weights across diverse environments. A more fundamental path is to directly generating the complete model weights $\theta$ via modeling the conditional distribution $p(\theta|e)$ (Figure 1).

Inspired by treating model weights as a data modality, this work focuses on generating environment-specific model weights (Figure 1c). By explicitly modeling the joint distribution of environments and weights, this generative adaptation is fundamentally suited for data-scarce scenarios where finetuning is impractical. However, the challenge of generating model weights for physical dynamics tailored to specific environments lies in three points. First, model weights, interconnected by the network architecture, are inherently structured. Thus, naive flattening weights into sequences would lead to the loss of crucial structural relationships (Kofinas et al., 2024). Second, the high dimensionality of weights results in an exceptionally vast parameter space. Minor variations in the weights of even a single layer

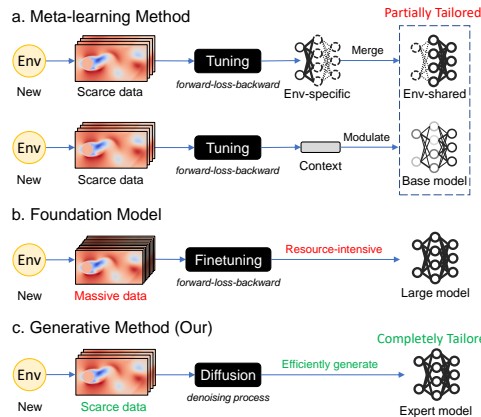

Figure 1: Paradigms for dynamics adaption.

can be amplified into significant difference in predictive performance (Plattner et al., 2025; Meynent et al., 2025). Therefore, traditional metrics like MSE are inadequate for assessing weight similarity. Finally, practical applications typically lack explicit physical knowledge of the environment, leaving only short trajectory snippets as available data. Consequently, it is necessary to extract discriminative features of the underlying dynamics from such limited observations.

To address these challenges, we propose a novel generative meta learning framework, Dynamics-informed weight Diffusion (DynaDiff). DynaDiff represents predictive models as weight graphs, aggregating weights into node features to preserve their inherent connectivity and accommodate arbitrary model architectures (challenge 1). It employs a node-attention Variational Autoencoder (VAE) to learn latent representations for the diffusion model, and incorporates a functional loss for weight similarity awareness (challenge 2). For unseen environments, we design a dynamics-informed prompter, which distills both physical features and temporal dynamics, thereby providing a highly informative prompt for the diffusion model (challenge 3). Finally, we propose a domain-adaptive model zoo that enables the efficient construction of a high-quality training corpus for DynaDiff.

Our contributions can be summarized as follows:

- We propose modeling the joint distribution of model weights on environments for cross-environment prediction, thereby rapidly generating expert weights for new environments without tuning.

- We construct weight graphs based on model architecture to preserve connectivity and design a functional loss for weight similarity perception. This significantly enhances the generative model's ability to learn effective representations of model weights.

- Extensive experiments on simulated and real-world systems demonstrate that DynaDiff improves cross-environment generalization, boosting average prediction accuracy by 10.78% over competitive baselines.

## 2 PRELIMINARY

### 2.1 PROBLEM DEFINITION

Given environmental conditions $e \in \mathcal{E}$, the time-dependent system dynamics function is instantiated as $\frac{dx}{dt} = f(x, t, e) = f_e(x, t) \in \mathcal{F}$. The environment space $\mathcal{E}$ and the function space $\mathcal{F}$ are linked by the governing equations $f$, forming a joint set $\{e, f_e\}$. We employ a data-driven model $f_{\theta,e}$, parameterized by $\theta$, to learn $f_e$, thereby formalizing the function space $\mathcal{F}$ as the model's weight space $\Theta$. The environment space is divided into an observed environment set $\mathcal{E}_{tr}$ and an unseen environment set $\mathcal{E}_{te}$, and consequently, the weight space is also partitioned into corresponding subspaces $\Theta_{tr}$ and $\Theta_{te}$. Treating model weights as the modeling object, we learn the inherent joint distribution of environments and weights from the joint observation space $\{\mathcal{E}_{tr}, \Theta_{tr}\}$. For a new environment $e \in \mathcal{E}_{te}$, we generate a corresponding predictive function $f_{\theta,e}$ once learning is complete.

Notably, we posit that even when sharing the same governing equations, each environment determines a unique dynamical function. At test time, given a short observation sequence $X_L = \{x_0, ..., x_{L-1}\}$

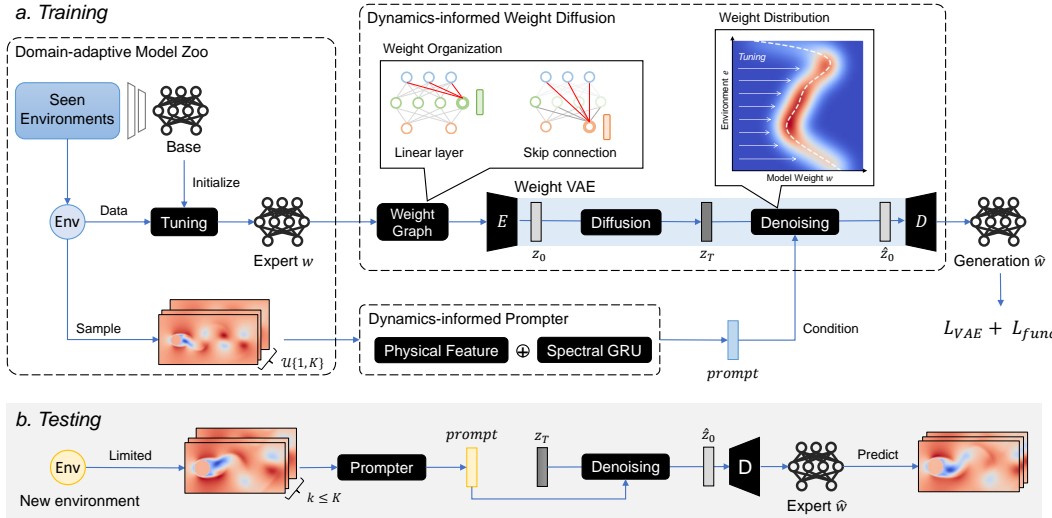

Figure 2: Framework of our Dynamics-informed weight Diffusion.

from a new, unseen environment $e \in \mathcal{E}_{te}$, our goal is to generate the complete model weights $\theta_{new}$ by modeling the conditional distribution $P(\theta|X_L)$. This approach, which generates a full expert model from scratch without requiring gradient-based finetuning, significantly differs from existing practices in dynamics prediction.

## 2.2 CONDITIONAL DIFFUSION

Diffusion models (Rombach et al., 2022; Dhariwal & Nichol, 2021) learn a probabilistic transformation from a prior Gaussian $p_{prior} \in \mathcal{N}(\mathbf{0}, \mathbf{I})$ distribution to a target distribution $p_{target}$. It perturbs data distributions by adding noise and learn to reverse this process through denoising, demonstrating strong fitting capabilities for data across modalities like images, language, and speech (Croitoru et al., 2023; Tumanyan et al., 2023). We denote the original diffusion sample as $x_0$. The forward noising process in standard diffusion models is computed as $x_n = \sqrt{\bar{a}_n} x_0 + \sqrt{1 - \bar{a}_n}\epsilon$, where $\epsilon$ and $\{\bar{a}_n\}$ represent the Gaussian noise and noise schedule (Ho et al., 2020), respectively. The reverse process gradually denoises from Gaussian noise to sample data as

$$p_\theta(x_{n-1}|x_n) := \mathcal{N}(x_{n-1}; \mu_\theta(x_n, n), \sigma_n^2 \mathbf{I}), \tag{1}$$

where $\mu_\theta = \frac{1}{\sqrt{\alpha_n}}(x_n - \frac{1 - \alpha_n}{\sqrt{1 - \bar{\alpha}_n}}\epsilon_\theta(x_n, n))$ and $\{\sigma_n\}$ are step dependent constants. The noise $\epsilon_\theta$ is computed by a parameterized neural network, typically implemented as a UNet or Transformer architecture. The network's parameters are optimized through an objective function (Ho et al., 2020)

$$L_n = \mathbb{E}_{n, \epsilon_n, x_0} ||\epsilon_n - \epsilon_\theta(\sqrt{\bar{\alpha}_n} x_0 + \sqrt{1 - \bar{\alpha}_n}\epsilon_n, n)||^2 \tag{2}$$

to minimize the negative log-likelihood $\mathbb{E}_{x_0 \sim q(x_0)}[-p_\theta(x_0)]$. To model conditional distributions $p(x|c)$, state-of-the-art methods inject conditional information during noise prediction using techniques like adaptive layer normalization (Peebles & Xie, 2023), as $\epsilon_\theta(x_n, n, c)$.

## 3 METHODOLOGY

In this section, we first introduce the method for modeling the joint distribution of model weights and environments, as illustrated in Figure 2a. Subsequently, we detail the efficient construction of a domain-adaptive model zoo. Finally, we present a dynamics-informed prompter that operates with the limited observation sequence.

### 3.1 DYNAMICS-INFORMED WEIGHT DIFFUSION

DynaDiff first organizes the expert model weights into a weight graph. It then pretrains a weight VAE, yielding a high-quality latent space. Finally, an dynamics-informed diffusion model is trained within this latent space.

### 3.1.1 WEIGHT GRAPH

Model weights constitute a novel data modality, inherently structured by the network architecture. A straightforward approach is to break the network structure and flatten weights layer by layer into fixed-length token sequences for representation using sequence models like transformers. Here, we consider the inherent connection structure of the neural network. Specifically, we aggregate layer weights based on the forward data flow through the network topology to construct a weight graph that encapsulates the network's connection structure.

We focus on designing the weight organization method for the basic computational units of modern AI architectures: linear layers and convolution layers (Kofinas et al., 2024). For a linear layer, learnable parameters include weights $\mathbf{w} \in \mathbb{R}^{D_{out} \times D_{in} \times 1}$ and bias $b \in \mathbb{R}^{D_{out} \times 1}$, where the $D_{out}$ and $D_{in}$ are the dimension of output and input respectively. A convolution layer similarly comprises weights $\mathbf{w} \in \mathbb{R}^{C_{out} \times C_{in} \times h \times w}$ and bias $b \in \mathbb{R}^{C_{out} \times 1}$, where $c_{out}$ and $c_{in}$ are the channels of output and input, respectively, and $h \times w$ is kernel size. We treat the output neurons of linear layers and output channels of convolution layers as nodes of

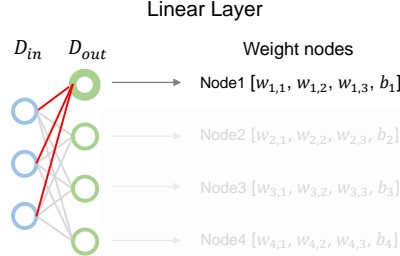

Figure 3: Weight node of linear layer.

the weight graph. Centering on the feature of output nodes, we flatten and concatenate the weights (and corresponding bias) associated with connections leading to each output node within a layer, forming the feature vector $\mathbf{w} \oplus b$ for that output node. Thus, a linear layer's weights are organized as $D_{out}$ nodes with $(D_{in} + 1)$-dimensional features (Figure 3), and a convolution layer's weights are organized as $C_{out}$ nodes with $(C_{in} \times h \times w + 1)$-dimensional features (Figure 8).

Considering the prevalence of skip connections in modern deep learning, we incorporate their weights. Following the data flow, we concatenate the weights of the skip connection path as additional features to the feature vector of the node where it merges with the main path, as depicted in Figure 8. Consequently, the entire model weights are structured as a weight graph with heterogeneous node features, where the total number of nodes equals the sum of the output neurons/channels across all layers. We normalize weights based on input-output node pairs and biases based on nodes.

The proposed weight graph aggregates weights to nodes. This not only captures inherent connection relationships but also significantly reduces computational overhead compared to maintaining dense edge features. This organization method is applicable to most architectures, as shown in Section 4.4.

### 3.1.2 WEIGHT VAE

We now encode the heterogeneous graph of model weights to build a low-dimensional and informative latent space for diffusion model. We train a node attention-based VAE with a loss function given by

$$L_{VAE} = -\mathbb{E}_{q_\phi(\mathbf{z}|\mathbf{w})}[\log p_\theta(\mathbf{w}|\mathbf{z})] + \beta\mathbf{KL}[q_\phi(\mathbf{z}|\mathbf{w})||p(\mathbf{z})], \qquad (3)$$

where $\mathbf{w}$ represents the heterogeneous node features of the weight graph, $\mathbf{z} \in \mathbb{R}^d$ is the latent representation, and the KL divergence term is used to constrain the posterior distribution $q_\phi(\mathbf{z}|\mathbf{w})$. The VAE architecture first employs a layer-wise linear mapping for each layer's nodes to project them into a same dimension. Subsequently, we utilize a multi-head attention mechanism to model inter-node relationships, capturing interactions among neurons within and across original model layers. The resulting latent representation $\mathbf{z} = E(\mathbf{w})$ is then passed through another layer-wise linear mapping, projecting it back to the original dimensions for reconstruction $\hat{\mathbf{w}} = D(\mathbf{z})$.

We notice that prediction models exhibiting similar performance can possess distinct parameter values (Meynent et al., 2025). This observation motivates our approach to the reconstruction error term in the VAE objective. The similarity between model weights should be gauged by their functional consistency, rather than merely their identical absolute values. We introduce a function loss,

$$L_{func} = \mathbb{E}_{x_i \in X}||f_{\hat{\mathbf{w}}}(x_i) - f_{\mathbf{w}}(x_i)||_2^2, \qquad (4)$$

where $f_{\mathbf{w}}(x_i)$ and $f_{\hat{\mathbf{w}}}(x_i)$ are the output values of the original and reconstructed weights, respectively, when applied to an input sample $x_i$. Intuitively, the function loss allows the VAE to reconstruct weights that may not appear identical to the originals but perform similarly. It relaxes the encoder's optimization constraints, promoting the learning of a latent space characterized by functional semantics. We theoretically analyze the effect of the function loss on generalization error in Appendix E.

### 3.1.3 Weight Latent Diffusion Model

In the latent space, we instantiate the noise network $\epsilon_\theta$ using a transformer architecture. Conditioned on dynamics-informed $prompt$, we inject this information into the network using adaptive layer norm (adaLN) (Peebles & Xie, 2023), forming $\epsilon_\theta(\mathbf{z}_n, n, prompt)$. Compared to performing diffusive generation directly on the raw weights (Yuan et al., 2024), the latent space offers significant dimensionality reduction, which alleviates the computationally intensive nature of the diffusion process and simplifies the generation of representations.

FIX

### 3.1.4 Dynamics-informed Prompter

In most practical scenarios, only a short observation sequence $X_L$ is available, instead of a known environmental parameter. The central task of the Prompter is to distill a rich, informative $prompt$ vector from this limited sequence $X_L$. To leverage the strengths of both domain knowledge and data-driven feature extraction, we design a hybrid architecture composed of two parallel branches.

First, we extract physical features to capture the system's macroscopic dynamics. For each state $x_i$, we compute its first and second-order moment statistics, energy, and enstrophy. For the resulting time series of length $L$ for each statistic, we then compute its temporal mean and trend to form the explicit prompt. Subsequently, we encode the microscopic evolution of the observation sequence. We compute the sequence of spectra for $X_L$ via Fast Fourier Transform (FFT), stacking the real and imaginary parts. A Gated Recurrent Unit (GRU) is then used to capture the evolutionary patterns across frames, with its final hidden state serving as the implicit prompt. We concatenate the explicit and implicit prompts to form the final dynamics-informed $prompt$. The computational details are provided in the Appendix F.

We sample observation sequences with a variable length ranging from $1$ to $L$ for each training epoch. This enables the prompter to handle a flexible number of observation frames at test time. Additionally, the prompter is trained jointly with the latent diffusion model, and its output is passed through an additional linear layer to regress on the ground-truth environmental condition $e$, with the regression error $L_{aux} = ||e - \mathrm{linear}(prompt)||_2^2$ serving as an auxiliary supervisory signal. We verify that this loss, although not necessary, helps to improve the interpretability of $prompt$ in the Appendix H.5.

FIX

## 3.2 Domain-adaptive Model Zoo

DynaDiff operates on expert model weights, which are collected in a pre-constructed model zoo. While a naive approach would be to train each expert model from scratch (Schürholt et al., 2024), this process is computationally prohibitive and leads to a non-stationary weight distribution. To address this, we introduce an efficient construction process centered on domain-adaptive initialization (Chen et al., 2024b). First, we pretrain a global base model on data from all visible environments, analogous to the environment-shared weights in meta-learning. Subsequently, each environment-specific expert is obtained by rapidly fine-tuning this base model, as illustrated in Figure 2. To encourage sufficient exploration of the weight landscape, we also introduce a minor random noise to one layer of the base model before each fine-tuning process. Since each expert only needs to solve for a specific environment, its size is substantially smaller than a general-purpose foundation model. Therefore, our model zoo trades affordable offline storage for a massive gain in training efficiency, eliminating the need for the inner-loop optimization common in prior meta-learning approaches (Finn et al., 2017; Dupont et al., 2022). Moreover, this one-time offline investment eliminates the need for any gradient-based computation when adapting to a new environment.

## 3.3 Generalization Analysis

We provide a theoretical analysis in Appendix E, demonstrating that our framework is principally designed to control its generalization error. First, by training a VAE with a functional loss , we construct a latent space that is functionally smooth, where proximity between latent vectors reflects the functional similarity of the decoded models. Next, a conditional diffusion model then accurately generate representations within this well-behaved space. Coupled with an auxiliary loss that grounds the prompter, this design ensures that each source of the total error is directly governed and minimized by a specific training objective.

Table 1: Average RMSE (± std from 5 runs) in in- and out-domain environments. Best in bold, underlined for suboptimal. The parameter sizes of predictive models are reported.

| Methods | Testing Params | Cylinder Flow (96:400) | | Lambda-Omega (12:39) | | Kolmogorov Flow (12:39) | | Navier-Stokes (24:121) | |
|---|---|---|---|---|---|---|---|---|---|
| | | In-domain | Out-domain | In-domain | Out-domain | In-domain | Out-domain | In-domain | Out-domain |
| Not-Adaptive | $\sim 1M$ | $0.124_{\pm 0.026}$ | $0.159_{\pm 0.029}$ | $0.214_{\pm 0.045}$ | $0.232_{\pm 0.042}$ | $0.135_{\pm 0.027}$ | $0.149_{\pm 0.029}$ | $0.129_{\pm 0.030}$ | $0.144_{\pm 0.033}$ |
| One-per-Env | | $0.040_{\pm 0.040}$ | $0.038_{\pm 0.040}$ | $0.038_{\pm 0.032}$ | $0.035_{\pm 0.008}$ | $0.069_{\pm 0.021}$ | $0.071_{\pm 0.019}$ | $0.046_{\pm 0.007}$ | $0.047_{\pm 0.009}$ |
| **One-for-All** FNO | $\sim 500M$ | $0.082_{\pm 0.025}$ | $0.083_{\pm 0.023}$ | $0.352_{\pm 0.041}$ | $0.363_{\pm 0.040}$ | $0.080_{\pm 0.020}$ | $0.096_{\pm 0.016}$ | $\underline{0.066}_{\pm 0.009}$ | $\underline{0.074}_{\pm 0.015}$ |
| DPOT | $\sim 500M$ | $0.091_{\pm 0.008}$ | $0.090_{\pm 0.007}$ | $0.324_{\pm 0.007}$ | $0.325_{\pm 0.007}$ | $\mathbf{0.079}_{\pm 0.012}$ | $\underline{0.084}_{\pm 0.017}$ | $0.087_{\pm 0.021}$ | $0.093_{\pm 0.020}$ |
| Poseidon | $\sim 600M$ | $0.085_{\pm 0.014}$ | $0.083_{\pm 0.015}$ | $0.301_{\pm 0.013}$ | $0.318_{\pm 0.009}$ | $0.102_{\pm 0.006}$ | $\underline{0.103}_{\pm 0.005}$ | $0.092_{\pm 0.017}$ | $0.095_{\pm 0.016}$ |
| MPP | $\sim 550M$ | $0.102_{\pm 0.020}$ | $0.098_{\pm 0.019}$ | $0.311_{\pm 0.054}$ | $0.313_{\pm 0.055}$ | $0.098_{\pm 0.017}$ | $0.103_{\pm 0.022}$ | $0.095_{\pm 0.026}$ | $0.096_{\pm 0.028}$ |
| **Env-Adaptive** DyAd | | $0.096_{\pm 0.021}$ | $0.094_{\pm 0.020}$ | $0.138_{\pm 0.078}$ | $0.137_{\pm 0.075}$ | $0.099_{\pm 0.006}$ | $0.098_{\pm 0.005}$ | $0.091_{\pm 0.018}$ | $0.096_{\pm 0.015}$ |
| LEADS | | $0.101_{\pm 0.031}$ | $0.115_{\pm 0.036}$ | $0.121_{\pm 0.031}$ | $0.123_{\pm 0.032}$ | $0.107_{\pm 0.011}$ | $0.105_{\pm 0.010}$ | $0.091_{\pm 0.022}$ | $0.094_{\pm 0.020}$ |
| CoDA | $\sim 1M$ | $0.099_{\pm 0.029}$ | $0.100_{\pm 0.031}$ | $0.119_{\pm 0.034}$ | $0.116_{\pm 0.032}$ | $0.097_{\pm 0.019}$ | $0.098_{\pm 0.019}$ | $0.096_{\pm 0.016}$ | $0.098_{\pm 0.014}$ |
| GEPS | | $0.079_{\pm 0.018}$ | $0.082_{\pm 0.020}$ | $0.094_{\pm 0.041}$ | $0.092_{\pm 0.039}$ | $0.089_{\pm 0.009}$ | $0.086_{\pm 0.008}$ | $0.098_{\pm 0.011}$ | $0.099_{\pm 0.010}$ |
| CAMEL | | $0.089_{\pm 0.018}$ | $0.094_{\pm 0.016}$ | $0.104_{\pm 0.018}$ | $0.103_{\pm 0.018}$ | $0.096_{\pm 0.013}$ | $0.101_{\pm 0.016}$ | $0.106_{\pm 0.018}$ | $0.109_{\pm 0.015}$ |
| DynaDiff | | $\mathbf{0.059}_{\pm 0.028}$ | $\mathbf{0.065}_{\pm 0.025}$ | $\mathbf{0.090}_{\pm 0.021}$ | $\mathbf{0.089}_{\pm 0.023}$ | $\underline{0.081}_{\pm 0.012}$ | $\mathbf{0.080}_{\pm 0.013}$ | $\mathbf{0.062}_{\pm 0.017}$ | $\mathbf{0.063}_{\pm 0.015}$ |

## 4 EXPERIMENT

### 4.1 EXPERIMENTAL SETUP

We consider unknown environmental conditions for all dynamical systems, training models solely on observed trajectories across diverse visible environments. Test environments are categorized as in-domain (seen during training, novel initial conditions) and out-domain (unseen environments) (Nzoyem et al., 2024). At test time, models autoregressively predict future states given a single initial frame. The prediction horizon is 100 steps for Cylinder Flow and Lambda-Omega, and 50 steps for Kolmogorov Flow and Navier-Stokes. We evaluate prediction quality using root mean square error (RMSE) and structural similarity index (SSIM). By default, the length of the observation sequence for new environments is $L = 10$.    FIX

**Baselines** We compare against two baseline categories: foundation models (One-for-All) and meta-learning approaches (Env-Adaptive). The foundation models are trained via empirical risk minimization (Ayed et al., 2019) on trajectories from all visible environments, including DPOT (Hao et al., 2024), Poseidon (Herde et al., 2024), and MPP (McCabe et al., 2024). The meta-learning methods learn environment-shared weights and update environment-specific contexts on observation sequences, including DyAd (Wang et al., 2022), LEADS (Yin et al., 2021), CoDA (Kirchmeyer et al., 2022), GEPS (Koupaï et al., 2024), and CAMEL (Blanke & Lelarge, 2024). Following existing work Blanke & Lelarge (2024), we enable zero-shot prediction by conditioning the hypernetwork on environmental conditions $e$, which assumes known ground-truth environmental conditions. Additionally, we assume all environments are visible and train a dedicated Fourier neural operator (Li et al., 2020) (FNO) for each environment as a performance upper bound (One-per-Env). We also train an FNO only on all visible environments, but test without any adaptation as a performance lower bound (Not-Adaptive). Unless otherwise specified, we use FNO as the expert small model for DynaDiff and other meta-learning methods. Detailed architectural are in Appendix C and I.    FIX

**Dynamical Systems** We validate the model's effectiveness on four time-dependent PDE systems and one real-world dataset: 1) Cylinder Flow (Li et al., 2025a); 2) Lambda-Omega (Champion et al., 2019); 3) Kolmogorov Flow (Koupaï et al., 2024); 4) Navier-Stokes Equations (Kirchmeyer et al., 2022); and 5) ERA5 Dataset (Zhang et al., 2025). For the PDE systems, we use equation coefficients or external forcing as environmental variables and simulate multiple trajectories under different environments for training and testing. We train 100 FNO weight sets for each seen environment across all systems to serve as the model zoo of DynaDiff (size 100). Detailed descriptions and data generation procedures for each system are provided in Appendix A and B.

### 4.2 MAIN RESULTS

**PDE systems** We report the generalization performance on 4 PDE systems in Table 1, detailing the number of in/out-domain environments and the parameter size of models for each system during testing. The generative module of DynaDiff has approximately 400M parameters, while the predictive

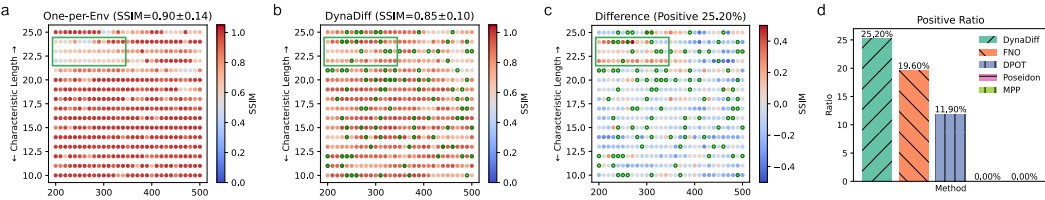

Figure 4: Predicting performance on Cylinder Flow. SSIM distribution of (a) One-per-Env and (b) DynaDiff; (c) Ratio where DynaDiff outperforms One-per-Env; (d) Differences between DynaDiff and One-per-Env. The green circle and box means seen environment during training and highlight region, respectively.

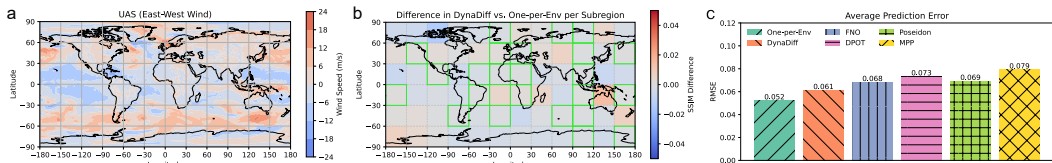

Figure 5: Predicting performance on ERA5 data. (a) One frame of ground true wind speed. (b) SSIM difference between DynaDiff and One-per-Env. The green box means seen environment during training. (c) Average prediction RMSE of DynaDiff and foundation models.

model at test-time has only 1M. Across nearly all systems, DynaDiff achieves the best average performance, demonstrating its ability to model the conditional dependence of the predictive model on environments. Its small, environment-specific expert models outperform foundation models with hundreds of times more parameters. Furthermore, unlike other meta-learning approaches, DynaDiff treats model weights holistically during adaption, without forcing the retention of environment-shared components. This potentially expands DynaDiff's search space for improved generalization.

We also find that some models can outperform One-per-Env in certain environments. This is likely due to the stochasticity of initialization and the training process, as One-per-Env models do not always converge to the optimal point. We illustrate this result with Cylinder Flow (2 environmental variables), as shown in Figure 4. The overall SSIM of One-per-Env is close to 1, however, it exhibits suboptimal performance in certain regions (green box in Figure 4). The FNO weights generated by DynaDiff perform better than One-per-Env in some environments, even unseen ones. This suggests that DynaDiff captures the manifold where the joint distribution of weights and environments lies, whereas the optimizer training process can fail to converge onto this manifold possibly due to getting stuck in local optima (Sclocchi & Wyart, 2024). We provide a further analysis of the weight manifold captured by DynaDiff in Appendix H.9.

**Real-world dataset**  We utilize the ERA5 reanalysis dataset, including east-west and north-south wind speed data at a height of 100 meters. The spatial resolution is 0.25°, and the temporal resolution is 1 hour. We use January 2018 wind speeds as the training set and January 2019 as the test set. To define different environments, we divide the globe into 6×12 grid subregions at 30° intervals (Wang et al., 2022). We randomly select 24 subregions as seen environments, with the remaining 48 as unseen environments. The experimental results are shown in Figure 5. DynaDiff's prediction performance outperform all baselines and is able to surpass One-per-Env in partial unseen subregions.

### 4.3 ROBUSTNESS

We investigate the impact of the number of seen environments, model zoo size and the length of observation sequence $L$. We first examine the effect of model zoo size on Cylinder Flow and Lambda-Omega systems, as depicted in Figures 9a and b. The results indicate that DynaDiff exhibits relatively stable performance with a zoo size of 50. As the zoo size decreases further, performance begins to deteriorate, even within the distribution. Subsequently, we test the influence of the number of seen environments on the Kolmogorov Flow and Navier-Stokes systems. The number of environments ranged from approximately 5% to 20% of the total. The findings reveal that increasing the number of

seen environments reduces prediction error, but the gains become marginal after reaching around 20%. This suggests that DynaDiff learns the underlying joint distribution of weights and environments from a small number of environments, rather than overfitting to trajectory samples within those environments. Finally, we test DynaDiff's sensitivity to the observation length $L$. The results in Table 5 show that DynaDiff robustly captures the dynamic context to generate suitable predictors even with fewer frames, a flexibility enabled by our variable-length training strategy (Section 3.1.3).

Furthermore, in Appendix H.2 and H.3, we evaluate two challenging generalization scenario with a highly skewed distribution of training and testing environments, where DynaDiff consistently outperforms all baselines.

## 4.4 EXTENSIBILITY

The weight graph structure proposed in Section 3.1.1 is capable of organizing neural networks of arbitrary architectures. Here, we extent to more neural operators as expert models within DynaDiff, including Wavelet Neural Operator (Tripura & Chakraborty, 2023) (WNO), and U-shape Neural Operator (Rahman et al., 2022) (UNO). Our experimental results on Cylinder Flow are presented in Figure 6. DynaD-

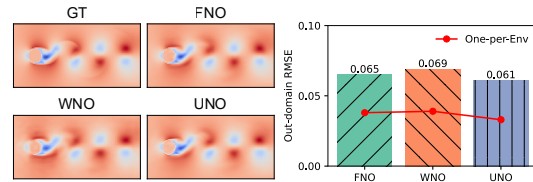

Figure 6: DynaDiff on the Cylinder Flow with different expert models of DynaDiff.

iff, when using different neural operators, consistently achieves excellent generalization performance, with actual performance showing only minor variations depending on the specific operator architecture. This demonstrates that DynaDiff is a model-agnostic framework capable of benefiting from the sophisticated architectural designs of its expert models. Detailed architectures of these neural operators are provided in Appendix I.

## 4.5 ABLATION STUDY

Here, we verify the necessity of domain initialization when building the model zoo and the function loss used during VAE training. Experimental results on the Kolmogorov Flow and Navier-Stokes systems are presented in Table 10. When function loss is omitted, the VAE relies solely on MSE for reconstruction similarity, leading to suboptimal generated weights, particularly in unseen environments. Function loss relaxes VAE encoding constraints and helps prevent overfitting by prioritizing functional consistency over exact reconstruction. Removing domain initialization results in a significant deterioration in generated weight performance. This is attributed to the high complexity of a randomly initialized model zoo, which significantly increases the modeling difficulty. We conclude that for weight generation aimed at generalization, sample quality is far more critical than diversity.

In addition, we compare the performance using our prompter against ground-truth environmental conditions, and analyze the impact of different diffusion architectures in Appendix H.6 and H.4.

## 4.6 COMPUTATIONAL COST

**Time cost** We compare the time overhead of DynaDiff and One-per-Env when adapting to new environments, as shown in Figure 7a. One-per-Env requires training weights for each new environment using observational data. DynaDiff's overhead includes building the model zoo (accelerated by domain initialization) and generating weights for new environments. Though the upfront time cost of preparing the model zoo, DynaDiff generates weights signifi-

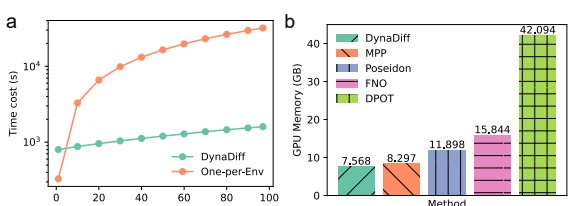

Figure 7: (a) Time cost and (b) GPU memory during testing on the Navier-Stokes system.

cantly faster than training a new predictor. This highlights the trade-off of our generative meta-learning paradigm: exchanging a moderate offline cost for significant test-time efficiency.

**GPU memory**   We compare the GPU memory usage of DynaDiff and other baselines during inference (Figure 7b). Thanks to the proposed weight graph structure, DynaDiff's attention computation unfolds along the node dimension, significantly reducing computational overhead and memory. In addition, we detail the storage overhead of the model zoo in Appendix B.

## 5   RELATED WORK

### 5.1   DYNAMICS PREDICTION ACROSS ENVIRONMENTS

Developing dynamic prediction models with cross-environment generalization is a crucial problem in scientific machine learning and has garnered significant research interest. We review the main approaches and related work in this area. The first category trains large-parameter neural solvers as foundation models using extensive simulated data (Rahman et al., 2024; Alkin et al., 2024; Chen et al., 2024a). Subramanian et al. (2023) explore the generalization performance of classical FNO architectures across different parameter sizes. Subsequently, models such as MPP (McCabe et al., 2024), DPOT (Hao et al., 2024), and Poseidon (Herde et al., 2024) employed more advanced architectures to improve computational efficiency and approximation capabilities. The second approach is meta-learning (Finn et al., 2017). These methods capture cross-environment invariants through environment-shared weights and fine-tune environment-specific weights or contexts on limited data from new environments for adaptation, including DyAd (Wang et al., 2022), LEADS (Yin et al., 2021), CoDA (Kirchmeyer et al., 2022), GEPS (Koupaï et al., 2024), CAMEL (Blanke & Lelarge, 2024), and NCF (Nzoyem et al., 2024). Additionally, other methods exist, like in-context learning (Chen et al., 2024b). Yang et al. (Yang et al., 2023) frame differential equation forward and inverse problems as natural language statements, pre-train transformers, and provide solution examples for new environments as context to enhance model performance. Compared to these works, we innovatively treat the complete model weights as generated objects and explicitly model their joint distribution with the environment.

### 5.2   DIFFUSION FOR NETWORK WEIGHT GENERATION

Generating neural network weights is a relatively nascent research area (Wang et al., 2024). An initial line of work involved training MLPs to overfit implicit neural fields, distilling them into model weights, and subsequently generating these MLP weights as an alternative to directly generating the fields (Erkoç et al., 2023; Li et al., 2025b). Another category proposes using generated weights to replace hand-crafted initialization, thereby accelerating and improving the neural network training process (Gong et al., 2024; Schürholt et al., 2022; 2024). These efforts primarily focus on image modalities. More recent studies leverage diffusion models to address generalization in various domains. Yuan et al. (Yuan et al., 2024) employ urban knowledge graph as prompts to guide diffusion for generating spatio-temporal prediction model weights for new cities. Zhang et al. (2024) replace the inner loop gradient updates of the meta learning with diffusion-generated weights. Xie et al. (2024) improve test-time generalization on time-varying systems by weight generation. Recent works (Soro et al., 2024; Charakorn et al., 2025; Liang et al., 2025) explores extracting features from unseen datasets and controlling diffusion to generate adapted model weights for them. However, most of these methods exhibit limited zero-shot performance. This may stem from them disrupting the neural network's inherent topological connections by directly flattening the weights, which constrains the representational capacity of the generative model. In contrast, we organize weights in the form of a neural graph and introduce a function loss to guide their representation. The prediction performance of our generated expert models without tuning surpasses larger pre-trained models.

## 6   CONCLUSION

We proposed DynaDiff, a framework for cross-environment generalization based on a new generative adaptation paradigm. DynaDiff synthesizes complete expert models from a few observations, guided by a dynamics-informed prompter and a generative model trained on a structured weight space. Our experiments demonstrate that this approach yields lightweight models with superior generalization, improving upon competitive baselines by an average of 10.78%. We conclude that generative weight modeling is a promising direction for scientific machine learning.

ETHICS STATEMENT

The authors acknowledge their responsibility to adhere to the ICLR Code of Ethics.

REPRODUCIBILITY STATEMENT

To ensure the reproducibility of our results, we provide comprehensive details of our methodology, experiments, and implementation.

- **Code and Data Availability.** We commit to making all source code, custom datasets, and data preprocessing scripts publicly available upon acceptance of this paper. The materials will be hosted in a public repository under a permissive open-source license to ensure full reproducibility and to facilitate future research.
- **Implementation Details.** A full description of our model architectures, algorithms, and experimental setup is provided in Appendix.

We believe this provides sufficient information for the research community to reproduce and build upon our findings.

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

IMPACT STATEMENT

This paper presents work whose goal is to advance the field of Machine Learning. There are many potential societal consequences of our work, none which we feel must be specifically highlighted here.

LIMITATIONS & FUTURE WORK

DynaDiff currently generates expert models of a fixed architecture, which may not be optimal for all possible environmental complexities. A promising future direction is to extend the generative paradigm to synthesize heterogeneous model architectures, dynamically tailored to each new environment.

THE USE OF LARGE LANGUAGE MODELS (LLMS)

We acknowledge the use of a large language model (LLM) to improving grammar and wording of our paper. The authors are fully responsible for the content of this work.

## A  DATA GENERATION

**Cylinder Flow system** (Li et al., 2025a) is governed by:

$$\begin{cases} \dot{u}_t = -u \cdot \nabla u - \dfrac{1}{\alpha}\nabla p + \dfrac{\beta}{\alpha}\Delta u, \\[2mm] \dot{v}_t = -v \cdot \nabla v + \dfrac{1}{\alpha}\nabla p - \dfrac{\beta}{\alpha}\Delta v. \end{cases} \tag{5}$$

In this system, we use the Reynolds number $Re$ and characteristic length $r$ as two environmental variables. The Reynolds number and characteristic length influence the lattice viscosity, which in turn affects the collision frequency, leading to different dynamic behaviors.

**Lambda–Omega system** (Champion et al., 2019) is governed by

$$\begin{cases} \dot{u}_t = \mu_u \Delta u + (1 - u^2 - v^2)u + \beta(u^2 + v^2)v \\[1mm] \dot{v}_t = \mu_v \Delta v + (1 - u^2 - v^2)v - \beta(u^2 + v^2)u, \end{cases} \tag{6}$$

where $\Delta$ is the Laplacian operator. For this system, we use $\beta$ as a 1-dimensional environmental variable. $\mu_v$ and $\mu_v$ are both set to 0.5.

**Kolmogrov Flow system** (Page et al., 2024) is governed by

$$\partial_t \omega + (\mathbf{u} \cdot \nabla)\omega = \frac{1}{Re}\Delta\omega - n\cos(ny),$$

$$\nabla^2 \psi = -\omega,$$

$$\mathbf{u} = (u, v) = \left(\frac{\partial \psi}{\partial y}, -\frac{\partial \psi}{\partial x}\right),$$

$$\omega = (\nabla \times \mathbf{u}) \cdot \hat{\mathbf{z}} = \frac{\partial v}{\partial x} - \frac{\partial u}{\partial y},$$

For this system, we use $Re$ as a 1-dimensional environmental variable. $n$ is set to 3.

**Navier-Stokes system** (Takamoto et al., 2022) is governed by

$$\frac{\partial \omega}{\partial t} + (\mathbf{u} \cdot \nabla)\omega = \nu\Delta\omega + f,$$

$$\nabla^2 \psi = -\omega,$$

$$\mathbf{u} = (u, v) = \left(\frac{\partial \psi}{\partial y}, -\frac{\partial \psi}{\partial x}\right),$$

$$\omega = (\nabla \times \mathbf{u}) \cdot \hat{\mathbf{z}} = \frac{\partial v}{\partial x} - \frac{\partial u}{\partial y},$$

where $f = A \left( \sin \left( 2\pi(x + y + s) \right) + \cos \left( 2\pi(x + y + s) \right) \right)$ is the driving force. We use amplitude $A$ and phase $s$ as the two-dimensional environmental variables for this system, and the viscosity coefficient is set to 0.01.

The range of environmental values and simulation settings for each equation are listed in Table 2.

The Cylinder flow system is simulated using the lattice Boltzmann method (LBM) (Vlachas et al., 2022), with dynamics governed by the Navier-Stokes equations for turbulent flow around a cylindrical obstacle. The system is discretized using a lattice velocity grid, and the relaxation time is determined based on the kinematic viscosity and Reynolds number. Data collection begins once the turbulence has stabilized.

For Lambda–Omega system, the system's reaction-diffusion equations are numerically integrated over time using an ODE solver.

For Kolmogorov Flow and Navier-Stokes systems, we perform numerical simulations based on the vorticity form equations. The process includes calculating the velocity field from vorticity by solving a Poisson equation using Fourier transforms, employing numerical methods to handle spatial derivatives, and subsequently using an ODE solver for time integration to simulate the evolution of vorticity over time.

Table 2: Simulation settings of each PDE system.

|  | Cylinder flow | Lambda–Omega | Kolmgorov Flow | Navier-Stokes |
|---|---|---|---|---|
| Spatial Domain | —— | $[-10, 10]^2$ | $[-\pi, \pi]^2$ | $[-32, 32]^2$ |
| Grid Num | $128 \times 64$ | $64 \times 64$ | $64 \times 64$ | $64 \times 64$ |
| dt | 200 | 0.04 | 0.2 | 0.025 |
| T | 45,000 | 40.0 | 40.0 | 50.0 |
| Environments | $Re : [200, 500, 31], r : [10, 25, 16]$ | $\beta : [1.0, 1.5, 51]$ | $Re : [50, 250, 51]$ | $A : [0.1, 0.3, 11], s : [0.0, 1.0, 11]$ |

For each environment of each system, we predict 100 trajectories from different initial conditions for training and 20 trajectories for testing. For Cylinder Flow and Lambda–Omega systems, autoregressive prediction is performed for 100 steps during testing, while for Kolmogorov Flow and Navier-Stokes systems, prediction is performed for 50 steps during testing.

## B  MODEL ZOO

In our main experiments, the settings for all meta-learning methods (except for `DyAd`, which uses a UNet by default) and the basic model of DynaDiff are shown in Table 3. Additionally, we report the storage overhead of the model zoo and the hyperparameter settings during generation. During training, we uniformly use the Adam optimizer with a learning rate of $1e - 4$, and other parameters are set to their default values.

Table 3: Detailed settings of the model zoo for each systems.

|  | Cylinder flow | Lambda–Omega | Kolmgorov Flow | Navier-Stokes | ERA5 |
|---|---|---|---|---|---|
| Channel Num | 2 | 2 | 3 | 3 | 2 |
| N_modes | $[12, 6]$ | $[8, 8]$ | $[8, 8]$ | $[8, 8]$ | $[8, 8]$ |
| N_layers | 4 | 4 | 8 | 8 | 4 |
| Hidden | 64 | 64 | 64 | 64 | 64 |
| Domain Pretraining (epochs) | 20 | 100 | 10 | 10 | 50 |
| Finetuning (epochs) | 50 | 50 | 50 | 50 | 20 |
| Storage Space (GB) | 18.0 | 3.5 | 3.5 | 7.1 | 3.4 |
| Time Cost per Model (s) | 6.7 | 28.6 | 5.15 | 56.28 | 30.47 |

## C  MODEL ARCHITECTURE

The learnable parameters of DynaDiff consist of a weight VAE and a weight latent transformer diffusion model. The VAE includes layer-wise linear projection layers at the start and end stages,

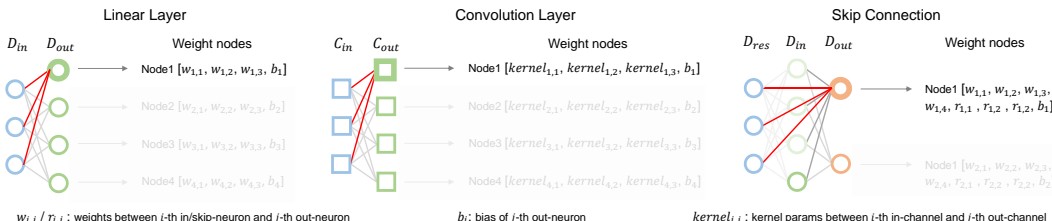

Figure 8: Layer-wise weight aggregation via forward data flow.

and inter-node attention layers in between. The diffusion model includes a noise network with a transformer architecture, where conditions are injected through adaptive layer normalization. The core model hyperparameters are configured as follows:

```
# --- VAE Hyperparameters ---
internal_dim = 1024      # Common internal dimension (D)
latent_dim = 512         # Latent dimension (h)
num_heads = 8            # Attention heads
num_attn_layers = 4      # Renamed from num_gnn_layers
# --- DiT Hyperparameters ---
hidden_size = 768        # Transformer hidden states
depth = 12              # Number of transformer blocks/layers
num_heads = 12          # Number of attention heads
```

Taking the FNO configuration of the NS system as an example, the VAE with the above settings has $193.16M$ parameters, and the DiT has $131.53M$ parameters.

## D BASELINE IMPLEMENTATION

The same training settings were used for all models, including training for 100 epochs using the Adam optimizer with a learning rate of 1e-4. Regarding the selection of foundation model parameters, we uniformly adjusted the embedding dimension, number of layers, and number of heads based on the dimensions suggested in the original papers to ensure comparable parameter counts for all models. For environment-adaptive models, we primarily used the default hyperparameters.

## E GENERALIZATION ERROR ANALYSIS

In this section, we provide a theoretical analysis of the generalization error for our proposed framework, DynaDiff. Our goal is to bound the expected functional error of a generated model $\hat{w}$ for a new, unseen environment, given a short observation sequence $X_L$. Let $w^*$ be the weights of an ideal expert model for this environment. The total generalization error can be expressed as $\mathcal{E}_{total} = \mathbb{E}[L_{func}(\hat{w}, w^*)]$, where the expectation is taken over the distribution of unseen environments and their corresponding observation sequences.

Our framework can be conceptualized as a composition of two modules: a prompter $P$, which maps an observation sequence to a conditional prompt, $prompt = P(X_L)$, and a generator $G$, which maps the prompt to the final model weights, $\hat{w} = G(prompt)$. The generator $G$ itself is a composition of the latent diffusion model and the VAE decoder $D$. The total error arises from imperfections in both of these modules.

To formalize the analysis, we introduce the concept of an oracle prompt, $prompt^*$, which perfectly encapsulates all necessary information about the new environment. The total error can then be decomposed using the triangle inequality:

$$\sqrt{\mathcal{E}_{total}} \leq \sqrt{\mathbb{E}[L_{func}(G(prompt), G(prompt^*))]} + \sqrt{\mathbb{E}[L_{func}(G(prompt^*), w^*)]} \quad (7)$$

This decomposition separates the total error into two terms: the error induced by the imperfect prompter, and the inherent error of the generator even when given a perfect prompt.

**Bounding the Prompter-induced Error.** We first analyze the error originating from the prompter. We posit a key assumption that the generative process is smooth with respect to its conditioning. Specifically, we assume the generator $G$ is $\gamma$-Lipschitz continuous in its functional output space with respect to the prompt.

*Assumption 1 (Functional Lipschitz Continuity).* There exists a constant $\gamma > 0$ such that for any two prompts, $prompt_1$ and $prompt_2$, the following holds:

$$\mathbb{E}[L_{func}(G(prompt_1), G(prompt_2))] \leq \gamma \cdot ||prompt_1 - prompt_2||_2^2 \tag{8}$$

This assumption is encouraged by the smooth nature of the denoising process in diffusion models (Preechakul et al., 2022). Given this, the prompter-induced error is bounded by the prompter's own generalization error, $\mathcal{E}_{prompt} = \mathbb{E}[||P(X_L) - prompt^*||_2^2]$. In our framework, we provide direct supervision to the prompter via an auxiliary regression loss, $\mathcal{L}_{aux} = ||e - \text{linear}(prompt)||_2^2$, which forces the prompt to contain physically meaningful information correlated with the ground-truth environment $e$, thereby helping to minimize $\mathcal{E}_{prompt}$.

**Bounding the Inherent Generator Error.** The second term represents the generator's error even under ideal conditioning. This error can be understood through the lens of domain adaptation theory (Redko et al., 2017; Wang et al., 2022), where the model zoo serves as the source domain and the unseen environments constitute the target domain. This error is primarily bounded by the generator's empirical performance on the model zoo, which we denote as $\mathcal{E}_{empirical}(G) = \mathbb{E}_{w \sim \Theta_{tr}}[L_{func}(G(prompt_w), w)]$, where $prompt_w$ is the prompt corresponding to an expert model $w$.

To demonstrate that this empirical error is itself bounded, we analyze the two-stage generative process. *Assumption 2 (Latent Diffusion Effectiveness).* An effectively trained conditional diffusion model can reverse the noising process in the latent space with high fidelity. This implies that the expected reconstruction error in the latent space is small. Let $z = E(w)$ be the latent representation of an expert model $w$. The expected squared error between $z$ and its reconstruction $\hat{z}$ after the full forward-and-reverse diffusion process is bounded by a small constant $\epsilon_z$:

$$\mathbb{E}[||z - \hat{z}||_2^2] \leq \epsilon_z \tag{9}$$

This is a standard assumption, as minimizing the denoising objective $L_n$ (Eq. 2) directly optimizes for this reconstruction capability. *Assumption 3 (Functional Smoothness of VAE Decoder).* The VAE decoder $D$ learns a smooth mapping from the latent space back to the functional space. This is a direct consequence of incorporating the functional loss $L_{func}$ (Eq. 4) during its training. This implies the decoder is $L_D$-Lipschitz continuous in a functional sense:

$$\mathbb{E}[L_{func}(D(z_1), D(z_2))] \leq L_D \cdot ||z_1 - z_2||_2^2 \tag{10}$$

The functional loss explicitly regularizes the mapping to ensure that small perturbations in the latent space do not lead to drastic changes in model behavior, thus encouraging a small $L_D$.

With these assumptions, we can bound the generator's empirical error. Using the triangle inequality on the square root of the functional loss:

$$\sqrt{\mathcal{E}_{empirical}(G)} = \sqrt{\mathbb{E}[L_{func}(D(\hat{z}), w)]} \leq \sqrt{\mathbb{E}[L_{func}(D(\hat{z}), D(z))]} + \sqrt{\mathbb{E}[L_{func}(D(z), w)]} \tag{11}$$

The first term on the right-hand side is the error from latent space generation, bounded by $\sqrt{L_D \cdot \epsilon_z}$ due to Assumptions 2 and 3. The second term is precisely the VAE's functional reconstruction error on the training data, which is minimized by the $L_{func}$ term in the VAE objective (Eq. 4). Let us denote the value of this minimized loss as $\epsilon_{recon}$.

This yields the final bound on the generator's empirical error:

$$\mathcal{E}_{empirical}(G) \leq (\sqrt{L_D \cdot \epsilon_z} + \sqrt{\epsilon_{recon}})^2 \tag{12}$$

This inequality demonstrates that the generator's performance on the training data is directly controlled by two terms that are actively minimized during our training procedure: the VAE's functional reconstruction loss and the diffusion model's denoising loss. This provides a strong theoretical justification for the stability and effectiveness of our framework.

## F    COMPUTATIONAL DETAILS OF THE DYNAMICS-INFORMED PROMPTER

Here, we detail the computational procedure for the Dynamics-informed Prompter module. The prompter takes a short observation sequence $X_L = \{x_0, x_1, \ldots, x_{L-1}\}$ as input, where each state $x_i \in \mathbb{R}^{C \times H \times W}$ is a multi-channel spatial field. The entire sequence has a shape of $(L, C, H, W)$.

### F.1    EXPLICIT PHYSICAL FEATURE EXTRACTOR

This branch computes a set of macroscopic physical statistics to capture the global dynamics. Let $S_k \in \mathbb{R}^L$ be the time series for the $k$-th statistic.

**Instantaneous Statistics.**    For each frame $x_i$ in the sequence, we compute four statistics. After calculation, we sum the values over the channel dimension $C$ to obtain a scalar value for each frame.

- **Spatial Mean (1st Moment):** The average value over the spatial domain.

$$\mu(x_i) = \frac{1}{H \times W} \sum_{h,w} x_{i,:,h,w}$$

- **Spatial Variance (2nd Moment):** The variance over the spatial domain.

$$\sigma^2(x_i) = \frac{1}{H \times W} \sum_{h,w} (x_{i,:,h,w} - \mu(x_i))^2$$

- **Energy (L2 Norm Squared):** A proxy for the total energy of the system.

$$E(x_i) = \sum_{h,w} \|x_{i,:,h,w}\|_2^2$$

- **Enstrophy (Squared Gradient Norm):** A proxy for the energy in the smallest scales, indicating turbulence.

$$\Omega(x_i) = \sum_{h,w} \|\nabla x_{i,:,h,w}\|_2^2$$

**Temporal Aggregation.**    For each of the four statistic time series $S_k$ (where $k \in \{\mu, \sigma^2, E, \Omega\}$), we compute two features to summarize its temporal evolution:

- **Temporal Mean:** The average value of the statistic over the sequence length $L$.

$$\bar{S}_k = \frac{1}{L} \sum_{i=0}^{L-1} S_{k,i}$$

- **Temporal Trend:** A simple approximation of the overall trend, calculated as the difference between the last and first values.

$$\Delta S_k = \frac{1}{L}(S_{k,L-1} - S_{k,0})$$

The final explicit prompt, $p_{\text{explicit}}$, is formed by concatenating these features for all statistics. If we compute $N_{\text{stats}} = 4$ statistics, the resulting vector has a shape of $N_{\text{stats}} \times 2 = 8$.

$$p_{\text{explicit}} = [\bar{S}_\mu, \Delta S_\mu, \bar{S}_{\sigma^2}, \Delta S_{\sigma^2}, \bar{S}_E, \Delta S_E, \bar{S}_\Omega, \Delta S_\Omega] \in \mathbb{R}^8$$

### F.2    IMPLICIT SPATIOTEMPORAL ENCODER

This branch learns a latent representation of the microscopic dynamics from the raw data sequence.

**Spectral Transformation.**    Each frame $x_i$ is transformed into its frequency representation $s_i$ using a 2D Fast Fourier Transform (FFT).

$$s_i = \text{FFT}(x_i) \in \mathbb{C}^{C \times H \times W}$$

We then stack the real and imaginary parts of the complex-valued spectra to form a real-valued tensor $\hat{s}_i \in \mathbb{R}^{2C \times H \times W}$, which is then flattened into a vector.

**Temporal Encoding with GRU.** The sequence of flattened spectra vectors $\{\hat{s}_0, \hat{s}_1, \ldots, \hat{s}_{L-1}\}$ is fed into a Gated Recurrent Unit (GRU). The GRU iteratively updates its hidden state $h_i$ based on the current input $\hat{s}_i$ and the previous hidden state $h_{i-1}$:

$$h_i = \text{GRU}(\hat{s}_i, h_{i-1})$$

The final hidden state, $h_{L-1}$, which encapsulates the temporal evolution of the entire spectral sequence, is taken as the implicit prompt, $p_{\text{implicit}}$. If the GRU's hidden dimension is $D_{\text{hidden}}$, the shape of the implicit prompt is $D_{\text{hidden}}$.

$$p_{\text{implicit}} = h_{L-1} \in \mathbb{R}^{D_{\text{hidden}}}$$

### F.3 FINAL PROMPT CONCATENATION

The final dynamics-informed prompt $p$ is obtained by concatenating the explicit and implicit vectors.

$$p = \text{concat}(p_{\text{explicit}}, p_{\text{implicit}})$$

The resulting prompt vector has a shape of $(8 + D_{\text{hidden}})$. This vector serves as the condition for the diffusion model.

## G   COMPUTATIONAL DETAILS OF THE WEIGHT VAE

Here, we elaborate on the architecture of the "node attention-based VAE". This architecture consists    NEW
of a symmetric encoder $E$ and decoder $D$, designed to process the weight graph $W$ defined by heterogeneous node features. A key design choice is that this VAE does not use global pooling along the node dimension. Instead, it learns a dedicated latent variable for every node in the weight graph (i.e., each neuron or channel in the FNO model).

**Input** The input to the VAE is the weight graph $W$, which consists of $L$ node feature tensors from different layers (e.g., lifting, FNO blocks, projection), denoted as $W = \{W_1, \ldots, W_L\}$.

- $W_i \in \mathbb{R}^{B \times N_i \times D_i}$ is the node feature tensor for the $i$-th layer.
- $B$ is the batch size.
- $N_i$ is the number of nodes in the $i$-th layer (e.g., the number of output channels).
- $D_i$ is the original feature dimension of the nodes in the $i$-th layer (e.g., $D_i = (C_{\text{in}} \times k_h \times k_w + 1)$).
- The total number of nodes is $N_{\text{total}} = \sum_{i=1}^{L} N_i$.

**Encoder** $E(W) \to (\mu_z, \sigma_z^2)$ The encoder $E$ compresses the input heterogeneous weight graph $W$ into per-node Gaussian distribution parameters.

- Step 1: Node Projection. To handle the heterogeneous dimensions $D_i$, we first use a set of layer-specific linear maps, $\text{MLP}_{\text{enc},i}$, to project all node features into a uniform, homogeneous embedding dimension $d_{\text{model}}$:

$$H_i = \text{GELU}(\text{MLP}_{\text{enc},i}(W_i)) \quad \in \mathbb{R}^{B \times N_i \times d_{\text{model}}}$$

- Step 2: Graph Re-assembly. We concatenate all projected node tensors $H_i$ along the node dimension (dim=1) to form a unified tensor $H_{\text{unified}}$ containing all nodes in the graph:

$$H_{\text{unified}} = \text{Concat}[H_1, \ldots, H_L] \quad \in \mathbb{R}^{B \times N_{\text{total}} \times d_{\text{model}}}$$

- Step 3: Node Attention Blocks. $H_{\text{unified}}$ is then passed through $K$ standard Transformer encoder blocks to capture complex relationships between nodes. For the $k$-th Transformer block ($k = 1 \ldots K$), the input is $H^{(k-1)}$ (where $H^{(0)} = H_{\text{unified}}$), and the computation proceeds as follows:

– **QKV Computation:** The block uses standard Multi-Head Self-Attention (MHA). The Query, Key, and Value are all derived from the same normalized input tensor $H_{\text{norm1}}^{(k)}$:

$$H_{\text{norm1}}^{(k)} = \text{LayerNorm}(H^{(k-1)})$$

$$Q^{(k)}, K^{(k)}, V^{(k)} \text{ are all derived from } H_{\text{norm1}}^{(k)}$$

– **Attention and Feedforward:**

$$H_{\text{attn}}^{(k)} = \text{MHA}(H_{\text{norm1}}^{(k)}, H_{\text{norm1}}^{(k)}, H_{\text{norm1}}^{(k)})$$

$$H_{\text{res1}}^{(k)} = H^{(k-1)} + H_{\text{attn}}^{(k)}$$

$$H_{\text{norm2}}^{(k)} = \text{LayerNorm}(H_{\text{res1}}^{(k)})$$

$$H_{\text{ffn}}^{(k)} = \text{FeedForward}(H_{\text{norm2}}^{(k)})$$

$$H^{(k)} = H_{\text{res1}}^{(k)} + H_{\text{ffn}}^{(k)}$$

After $K$ layers, we obtain the encoder output $H_{\text{enc\_out}} = H^{(K)}$.

- Step 4: Per-Node Latent Projection. $H_{\text{enc\_out}}$ is directly projected into the per-node latent variable parameter space (dimension $2 \cdot d_z$):

$$\text{Params}_{\text{latent}} = \text{MLP}_{\text{latent}}(H_{\text{enc\_out}}) \quad \in \mathbb{R}^{B \times N_{\text{total}} \times (2 \cdot d_z)}$$

Finally, we split this tensor along the last dimension to get the mean $\mu_z$ and log-variance $\log \sigma_z^2$:

$$\mu_z, \log \sigma_z^2 = \text{Split}(\text{Params}_{\text{latent}}) \quad \in \mathbb{R}^{B \times N_{\text{total}} \times d_z}$$

**Reparameterization**   We use the standard reparameterization trick, sampling on a per-node basis:

$$\sigma_z = \exp(0.5 \cdot \log \sigma_z^2)$$

$$\epsilon \sim \mathcal{N}(0, I) \quad (\text{with shape } \mathbb{R}^{B \times N_{\text{total}} \times d_z})$$

$$z = \mu_z + \epsilon \cdot \sigma_z \quad \in \mathbb{R}^{B \times N_{\text{total}} \times d_z}$$

**Decoder** $D(z) \to \hat{W}$   The decoder $D$ has a symmetric structure to the encoder.

- Step 1: Decoder Latent Projection. The per-node latent variable $z$ is first projected back to the $d_{\text{model}}$ dimension:

$$H_{\text{dec\_in}} = \text{MLP}_{\text{dec\_latent}}(z) \quad \in \mathbb{R}^{B \times N_{\text{total}} \times d_{\text{model}}}$$

- Step 2: Decoder Attention Blocks. $H_{\text{dec\_in}}$ is then passed through $K$ Transformer decoder blocks (with independent weights). The computation is identical to the encoder's Step 3, ultimately producing the decoder output $H_{\text{dec\_out}} \in \mathbb{R}^{B \times N_{\text{total}} \times d_{\text{model}}}$.

- Step 3: Split and Inverse Projection. $H_{\text{dec\_out}}$ is first split back into $L$ tensors corresponding to the original layers, $H_{\text{dec\_out},i} \in \mathbb{R}^{B \times N_i \times d_{\text{model}}}$. Then, each tensor is mapped back to its original, heterogeneous dimension $D_i$ via its layer-specific inverse projection $\text{MLP}_{\text{dec},i}$:

$$\hat{W}_i = \text{MLP}_{\text{dec},i}(H_{\text{dec\_out},i}) \quad \in \mathbb{R}^{B \times N_i \times D_i}$$

The final reconstructed weight graph $\hat{W} = \{\hat{W}_1, \ldots, \hat{W}_L\}$ matches the dimensions of the input $W$.

# H   ADDITIONAL RESULTS

In this section, we provide additional experiments to further validate the robustness, generalization capabilities and design choices of our DynaDiff framework.

Table 4: $p$-values for the statistical significance test (Welch's t-test) of the RMSE difference between DynaDiff and the best-performing baseline from Table 1.

| Cylinder Flow (96:400) | | Lambda-Omega (12:39) | | Kolmogorov Flow (12:39) | | Navier-Stokes (24:121) | |
|---|---|---|---|---|---|---|---|
| In-domain | Out-domain | In-domain | Out-domain | In-domain | Out-domain | In-domain | Out-domain |
| $3.26 \times 10^{-9}$ | $2.90 \times 10^{-27}$ | 0.67 | 0.70 | 0.89 | 0.066 | 0.022 | $3.79 \times 10^{-18}$ |

Table 5: Average out-domain RMSE of various observation length $L$.

| $L$ | 2 | 4 | 6 | 8 | 10 |
|---|---|---|---|---|---|
| Cylinder Flow | $0.069_{\pm 0.032}$ | $0.068_{\pm 0.034}$ | $0.064_{\pm 0.034}$ | $0.062_{\pm 0.031}$ | $0.059_{\pm 0.028}$ |
| Navier-Stokes | $0.080_{\pm 0.034}$ | $0.071_{\pm 0.033}$ | $0.068_{\pm 0.026}$ | $0.065_{\pm 0.023}$ | $0.062_{\pm 0.017}$ |

## H.1 STATISTICAL SIGNIFICANCE ANALYSIS OF MAIN RESULTS

We supplemented the key results in Table 1 with a statistical significance analysis, as suggested by the reviewer. We conducted pairwise Welch's t-tests on the RMSE scores between DynaDiff and the best-performing baseline (underlined in Table 1) for each environment. The $p$-values, computed using `scipy.stats.ttest_ind`, are presented in Table 4.  NEW

On the Cylinder Flow and Navier-Stokes systems, the improvements by DynaDiff are statistically significant ($p < 0.05$). In the Kolmogorov Flow system, although DPOT ($0.079 \pm 0.012$) performed slightly better on average than DynaDiff ($0.081 \pm 0.012$) in the in-domain setting, this difference is not statistically significant. However, in the more critical out-domain generalization task, DynaDiff's ($0.080 \pm 0.013$) advantage over GEPS ($0.084 \pm 0.017$) approaches statistical significance ($p = 0.066$). In the Lambda-Omega system, DynaDiff's advantage is not statistically prominent, indicating its performance is comparable to the SOTA baseline.

It is worth noting that DynaDiff consistently achieves strong generalization performance across almost all systems. Conversely, no single baseline demonstrates outstanding performance across all systems. This significantly indicates that DynaDiff, as a novel paradigm of weight-space learning, can generalize stably across different types of systems. This may be attributed to the fact that although the data for each PDE system varies greatly, DynaDiff models the weight distribution of a uniformly structured predictive model (e.g., FNO), making it more robust to dataset-level shifts. This cross-scenario stability highlights the superiority of DynaDiff's weight-space learning paradigm.

## H.2 ROBUSTNESS TO ENVIRONMENTAL EXTRAPOLATION

Standard out-of-domain tests often involve interpolating between seen parameter values. A more challenging test is extrapolation, where the model must predict system behavior in a region of the parameter space far from the training data.

We conducted a difficult extrapolation experiment on the Cylinder Flow system, which is governed by two environmental parameters. We constructed a biased training set containing only environments from the top-right quadrant of the parameter space (i.e., where both parameters had high values). The model was then tested on the unseen bottom-left quadrant (i.e., where both parameters had low values).

The results are summarized in Table 6. As expected, this task is extremely challenging for all methods, and performance degrades as the training distribution becomes more biased (i.e., the seen environment ratio decreases). However, our method, DynaDiff, consistently maintains a significant performance advantage over the strong baseline models. This demonstrates that by learning a coherent representation of the weight-environment manifold, DynaDiff is more robust to extrapolation and less prone to catastrophic failure when faced with significant distributional shifts.

## H.3 GENERALIZATION TO UNSEEN GOVERNING EQUATIONS

To rigorously test the upper limits of our framework's generalization ability, we designed a challenging experiment where the model must generalize to a completely unseen physical system with different governing equations.

Table 6: Out-of-domain RMSE on the Cylinder Flow extrapolation task. The models were trained on a biased (top-right quadrant) subset of environments and tested on the unseen opposite quadrant.

| Seen Env. Ratio | 100% | 90% | 80% | 70% | 60% | 50% |
|---|---|---|---|---|---|---|
| Poseidon | 0.083 | 0.098 | 0.128 | 0.214 | 0.568 | 0.674 |
| GEPS | 0.093 | 0.126 | 0.136 | 0.143 | 0.183 | 0.654 |
| **DynaDiff (Ours)** | **0.065** | **0.077** | **0.095** | **0.107** | **0.121** | **0.228** |

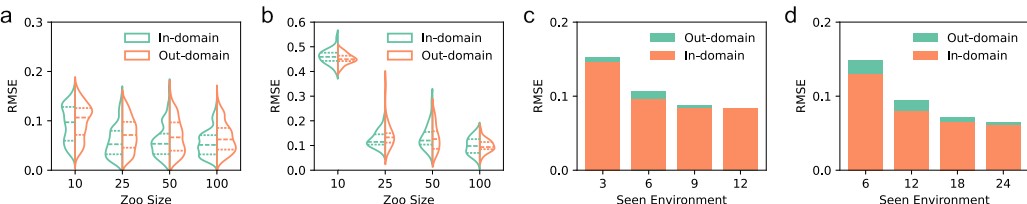

Figure 9: Robustness experiments. Impact of model zoo size on DynaDiff's performance on (a) Cylinder Flow and (b) Lambda-Omega. Impact of the number of seen environments (e) on DynaDiff's performance on (c) Kolmogorov Flow and (d) Navier-Stokes.

We trained a single, unified generative model on three distinct PDE systems: Cylinder Flow, Lambda-Omega, and Navier-Stokes. The test was then performed on a completely held-out system: Kolmogorov Flow. To create a unified conditioning space for the prompter, we treated the combination of the PDE type and its specific physical coefficients as a single, high-dimensional environmental descriptor. When combining data from different systems with varying channel numbers, we padded the input channels with zero to maintain a consistent tensor shape.

We evaluated the performance of all methods in both zero-shot and few-shot settings. For a fine-grained comparison, we measured the average prediction length (number of autoregressive steps) for which the Structural Similarity (SSIM) index remains above 0.8. As shown in Table 7, our method demonstrates superior performance in both scenarios. In the zero-shot case, DynaDiff achieves the longest accurate prediction horizon. For the few-shot setting, where each model was fine-tuned on a single trajectory from the held-out system, DynaDiff still maintained its advantage, showcasing its ability to generate high-quality initial models that benefit more from minimal fine-tuning. This result suggests that our framework captures a more fundamental and transferable representation of dynamical systems, extending beyond simple parameter interpolation to the structure of the dynamics itself.

Table 7: Performance on the held-out Kolmogorov Flow system, measured by the average prediction length with SSIM > 0.8. DynaDiff demonstrates superior generalization to a completely unseen physical law.

| Method | Zero-shot | Few-shot (1 trajectory) |
|---|---|---|
| Poseidon | 12.7 | 33.2 |
| DPOT | 14.0 | 41.4 |
| MPP | 10.1 | 38.7 |
| **DynaDiff (Ours)** | **15.9** | **46.5** |

## H.4 ABLATION ON FRAMEWORK DESIGN CHOICES

Our framework is composed of a two-stage generative stack (VAE + Latent Diffusion) that operates on a graph-based representation of weights. Here, we provide ablation studies to justify these key design choices against simpler alternatives.

**Two-Stage vs. Single-Stage Generation.** One could bypass the VAE and train a conditional diffusion model directly on the weight graphs. We compare our two-stage approach against such a single-stage, graph-structured conditional diffusion baseline (same to our Graph VAE's architecture).

As shown in Table 8, our method's superior performance highlights the advantage of our design. The VAE first learns a semantically rich and low-dimensional manifold, which makes the subsequent generation task for the diffusion model more tractable and effective. This decoupling of representation learning from generation is crucial.

**Graph vs. Sequence Representation.** An alternative to our weight graph is to flatten the weights into a sequence and use a powerful sequence model like a Transformer. We compare our graph-based VAE against a sequence-based Transformer VAE. The results in Table 9 show that our graph-based approach is significantly more effective and parameter-efficient. By explicitly injecting the network's architectural prior, the graph representation provides a much stronger and more suitable inductive bias for this task compared to relying on positional embeddings in a sequence.

Table 8: Ablation on the generative stack. Our two-stage (VAE + Latent Diffusion) approach significantly outperforms a direct, single-stage graph diffusion model.

| Method | Cylinder Flow (RMSE) | Lambda-Omega (RMSE) |
|---|---|---|
| Single-Stage Graph Diffusion | 0.112 | 0.238 |
| **Two-Stage (Ours)** | **0.065** | **0.089** |

Table 9: Ablation on weight representation. Our graph-based approach is more effective and parameter-efficient than a sequence-based Transformer approach.

| Representation | VAE Params | CF (RMSE) | LO (RMSE) | KF (RMSE) | NS (RMSE) |
|---|---|---|---|---|---|
| Sequence-based | $\sim$1200M | 0.129 | 0.208 | 0.152 | 0.143 |
| **Graph-based (Ours)** | $\sim$380M | **0.065** | **0.089** | **0.080** | **0.063** |

Table 10: Average RMSE of ablation study on domain initialization and function loss. 'w/o' stands for 'without'.

| | Kolmgorov Flow | | Navier-Stokes | |
|---|---|---|---|---|
| | In-domain | Out-domain | In-domain | Out-domain |
| w/o Domain Init | $0.156_{\pm 0.082}$ | $0.188_{\pm 0.102}$ | $0.197_{\pm 0.0102}$ | $0.201_{\pm 0.098}$ |
| w/o Function Loss | $0.098_{\pm 0.034}$ | $0.104_{\pm 0.038}$ | $0.104_{\pm 0.045}$ | $0.110_{\pm 0.046}$ |
| DynaDiff | $\mathbf{0.081_{\pm 0.023}}$ | $\mathbf{0.080_{\pm 0.013}}$ | $\mathbf{0.062_{0.017}}$ | $\mathbf{0.063_{\pm 0.015}}$ |

## H.5 Ablation on Auxiliary Supervisory Signal

As mentioned in Section 3.1.3, the prompter's training utilizes an additional linear layer and a regression loss $L_{aux}$ as an auxiliary supervisory task. However, in many practical scenarios, the environmental condition $e$ is often unknown. Therefore, $L_{aux}$ cannot always be computed. Here, we conduct an ablation study on $L_{aux}$ using the Cylinder Flow and Lambda-Omega systems to examine whether the generative task of the diffusion model alone can ensure the prompter learns to capture dynamic information from observation frames. The experimental results are shown in Table 11. We find that removing $L_{aux}$ does not significantly impair DynaDiff's generalization capability. This strongly demonstrates that DynaDiff's core generalization ability primarily stems from the dynamical information extracted from the observation sequence $X_L$, rather than a dependency on the ground-truth environment $e$. The role of $L_{aux}$ is essentially to introduce a beneficial learning bias to guide training, without providing extra knowledge. With or without $L_{aux}$, DynaDiff acquires physical information through the $L$ observation frames. We also find that without $L_{aux}$, the regression performance of the dynamics-informed *prompt* learned by the Prompter on the true physical coefficients decreases for the Cylinder Flow system (Figure 11). This suggests that while $L_{aux}$ may not add extra dynamical information to the *prompt*, its constraint during training helps the prompter extract more interpretable

NEW

Table 11: Average RMSE of ablation study on $L_{aux}$.

| | Cylinder Flow | | Lambda-Omega | |
|---|---|---|---|---|
| | In-domain | Out-domain | In-domain | Out-domain |
| w/o $L_{aux}$ | $0.063_{\pm 0.023}$ | $0.064_{\pm 0.026}$ | $0.088_{\pm 0.027}$ | $0.091_{\pm 0.025}$ |
| DynaDiff | $0.059_{\pm 0.028}$ | $0.065_{\pm 0.025}$ | $0.090_{\pm 0.021}$ | $0.089_{\pm 0.023}$ |

dynamical representations. In summary, even when environmental conditions are unknown, DynaDiff can still reliably identify dynamical information from limited trajectory observations to generalize.

### H.6 PROMPTER

Here we conduct an experiment to validate the prompter's ability to capture physically meaningful information from limited observations. We perform this analysis on the Cylinder Flow and Lambda-Omega systems. Specifically, we use the dynamics-informed prompt extracted by the prompter to regress the ground-truth environmental coefficients using a Random Forest regressor. The target coefficients are the Reynolds number (Re) and characteristic length (r) for Cylinder Flow, and the coefficient Beta for the Lambda-Omega system. Figure 10 reports the regression performance on out-of-distribution environments. As shown, the predicted values correlate strongly with the true values, demonstrating that the prompter can reliably infer the physical parameters. This result indicates that the prompter has successfully learned to extract the underlying dynamic signature from limited observation frames. It can therefore encode a discriminative and physically-grounded prompt to effectively guide the diffusion-based weight generation.

We also compare the performance of DynaDiff when using real environmental conditions $e$ versus surrogate environmental labels $c$, as shown in Table 12. Experimental results indicate that there is little difference in DynaDiff's performance under the two settings. This demonstrates that the prompter effectively helps DynaDiff distinguish different environments for generating suitable weights.

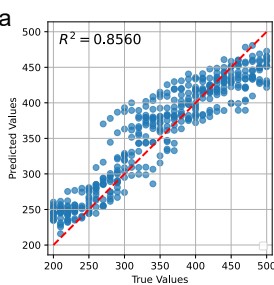 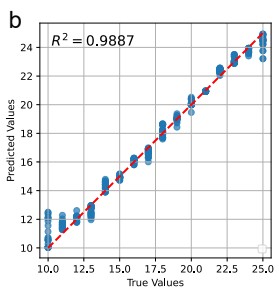 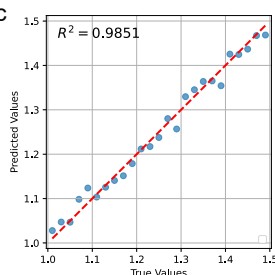

Figure 10: Prompter performance on the (a, b) Cylinder Flow and (c) Lambda-Omega systems.

 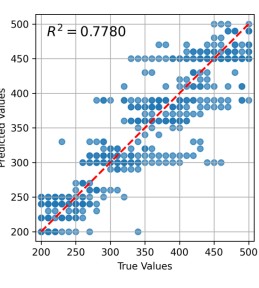 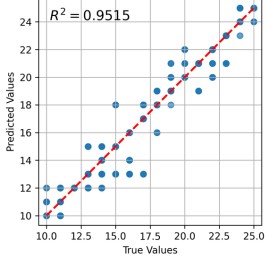 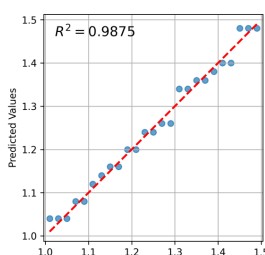

Figure 11: Prompter performance without $L_{aux}$ on the (a, b) Cylinder Flow and (c) Lambda-Omega systems.

Table 12: Average RMSE of ablation study on prompter.

| | Cylinder Flow | | Navier-Stokes | |
|---|---|---|---|---|
| | In-domain | Out-domain | In-domain | Out-domain |
| Prompter | $0.059_{\pm 0.028}$ | $0.065_{\pm 0.025}$ | $0.062_{\pm 0.017}$ | $\mathbf{0.063_{\pm 0.015}}$ |
| Environmental condition | $\mathbf{0.050_{\pm 0.014}}$ | $\mathbf{0.061_{\pm 0.027}}$ | $\mathbf{0.058 \pm 0.007}$ | $0.065_{\pm 0.006}$ |

Table 13: Time and memory costs of meta-learning methods on the Navier-Stokes system.

| | DyAd | LEADS | CoDA | GEPS | CAMEL | DynaDiff |
|---|---|---|---|---|---|---|
| GPU Memory (GB) | 0.796 | 0.902 | 1.122 | 1.028 | 0.846 | 2.168 |
| Time Cost (s) | 5.96 | 29.23 | 20.88 | 28.80 | 20.25 | 31.96 |

## H.7 TIME AND MEMORY COST OF META-LEARNING METHODS

We compare the inference cost of DynaDiff against all meta-learning methods for a single environment on the Navier-Stokes system, as shown in Table 13. Compared to other adaptive methods, the additional inference cost of DynaDiff comes from the generator, which includes latent space denoising and VAE decoding. The results show that, thanks to the low dimensionality of the latent space, the actual incremental cost is very small. Note that the GPU memory for DynaDiff reported in Figure 7b of the main paper was calculated during parallel inference across multiple environments, which is why it appears larger.

NEW

## H.8 ANALYSIS OF GENERATOR PARAMETER COUNT AND BASELINE SCALABILITY

The parameter count of DynaDiff's generator is approximately 400M, which is positioned between prior meta-learning methods (<50M) and foundation models (>500M). This difference in generator size is determined by the methodological paradigm: prior meta-learning approaches typically use a hypernetwork to generate low-dimensional context vectors, whereas our approach generates the complete weights of a 1M-parameter FNO model.

NEW

We conduct an experiment, using CAMEL and GEPS as representatives, where we increased their hypernetwork depth (3 layers) and width (15,000-dim) to scale them to a comparable size. The performance on the Cylinder Flow and Lambda-Omega systems is shown in Table 14. The results indicate that simply increasing the parameter count of the meta-learning baselines does not effectively improve their performance ceiling. This suggests that the performance bottleneck for these methods is not the parameter count itself. In contrast, our framework provides a new alternative that achieves higher generalization performance.

Table 14: RMSE performance of scaled-up meta-learning baselines vs. DynaDiff.

| | Cylinder Flow | | Lambda-Omega | |
|---|---|---|---|---|
| | In-domain | Out-domain | In-domain | Out-domain |
| GEPS-10M | 0.079 | 0.082 | 0.094 | 0.092 |
| GEPS-450M | 0.083 | 0.084 | 0.097 | 0.100 |
| CAMEL-5M | 0.089 | 0.094 | 0.104 | 0.103 |
| CAMEL-400M | 0.097 | 0.099 | 0.102 | 0.104 |
| DynaaDiff-400M | 0.059 | 0.065 | 0.090 | 0.089 |

## H.9 EXPLAINABILITY

We visualize the joint distribution of weights and environments using the Cylinder Flow system as an example to aid qualitative analysis, where over 80% environments are unseen by DynaDiff. In Figure 12, the x-axis represents the surrogate environment labels predicted by the prompter, and the y-axis represents the first principal component of the weights of a specific layer. The weight-environment landscape learned by DynaDiff closely resembles that learned by One-per-Env through

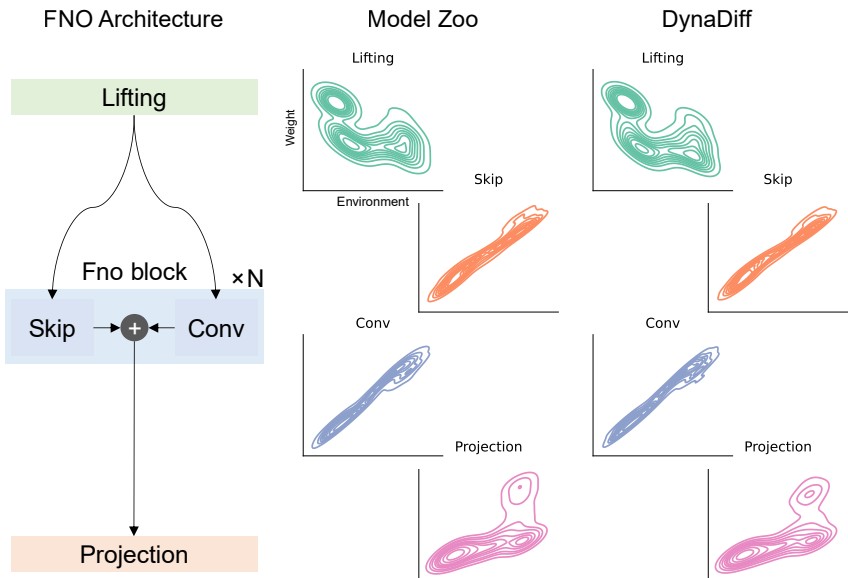

Figure 12: Joint distribution of weights and environments on Cylinder Flow.

optimizer training. This indicates that DynaDiff successfully models the joint distribution of weights and environments, thereby explaining its superior performance in Table 1.

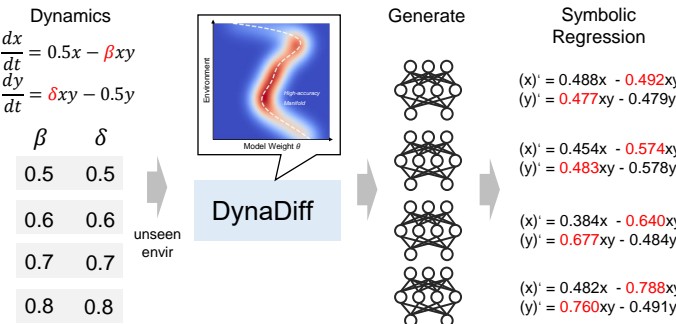

Figure 13: Generative functions of DynaDiff for the LV system.

To quantitatively analyze the environmental-weight joint distribution fitted by DynaDiff, we introduce a simple ODE system, the Lotka-Volterra (LV) equations (Kirchmeyer et al., 2022), as a toy example. We use a symbolic regression algorithm (Brunton et al., 2016) to distill the predictive model generated by DynaDiff for specific environmental conditions ($\beta$ and $\delta$) into an equation expression, as shown in Figure 13. The equivalent equation for the weights generated by DynaDiff is consistent in form with the LV equations, and the environmental coefficients are close. This quantitatively demonstrates that DynaDiff can fit the generalizable dynamics function rather than an environment-specific function. We detail the experimental setup in Appendix H.10.

### H.10 LV SYSTEM

The Lotka-Volterra equations describe the interaction between a predator-prey pair in an ecosystem:

$$\frac{dx}{dt} = \alpha x - \beta xy$$
$$\frac{dy}{dt} = \delta xy - \gamma y,$$

where $x$ and $y$ respectively represent the quantity of the prey and the predator, and $\alpha$, $\beta$, $\delta$, $\gamma$ define the species interactions. We generate each trajectory with a time interval of 0.1 and a total duration of 10.0, with initial conditions randomly sampled between 1 and 3. We change $\beta$ and $\delta$ as 2-dimensional environmental conditions, with both ranging from 0.5 to 1.0, sampled at 11 equally spaced points. Among a total of $11 \times 11 = 121$ environments, we randomly select 24 as the training set and the rest as the test set. For all environments, $\alpha = 0.5$ and $\gamma = 0.5$. We generate 100 trajectories for each environment.

For this system, we adopt a 2-layer MLP as the parameterized dynamic model, with a hidden layer dimension of 128 and Tanh as the activation function. We use a neural ordinary differential equation with the *rk4* algorithm to model the dynamics. DynaDiff generates the weights of the MLP and makes forward predictions through a numerical solver. When distilling the generated weights, we first perform autoregressive prediction on a given trajectory using the predicted weights. Once the prediction is complete, we employ the pysindy library (Kaptanoglu et al., 2022; de Silva et al., 2020) for symbolic regression. The operator dictionary uses a 2nd order polynomial dictionary, and other hyperparameters are set to their default values.

## I   ARCHITECTURES OF EXPERT MODELS

In our main experiments, we deploy three neural operators as expert models for DynaDiff: FNO, UNO, and WNO. Here, taking the Cylinder Flow system as an example, we list the parameter composition and hyperparameter settings of these operators.

**FNO**   We adopt the code from the open-source repository (Kossaifi et al., 2024) as the implementation for FNO. For the NS system, the weight composition of FNO is as follows:

```
lifting.fcs.0: torch.Size([128, 5])
lifting.fcs.1: torch.Size([64, 129])
fno_blocks.convs.0: torch.Size([64, 6209])
fno_blocks.channel_mlp.0.0: torch.Size([32, 65])
fno_blocks.channel_mlp.0.1: torch.Size([64, 34])
fno_blocks.convs.1: torch.Size([64, 6209])
fno_blocks.channel_mlp.1.0: torch.Size([32, 65])
fno_blocks.channel_mlp.1.1: torch.Size([64, 34])
fno_blocks.convs.2: torch.Size([64, 6209])
fno_blocks.channel_mlp.2.0: torch.Size([32, 65])
fno_blocks.channel_mlp.2.1: torch.Size([64, 34])
fno_blocks.convs.3: torch.Size([64, 6209])
fno_blocks.channel_mlp.3.0: torch.Size([32, 65])
fno_blocks.channel_mlp.3.1: torch.Size([64, 34])
projection.fcs.0: torch.Size([128, 65])
projection.fcs.1: torch.Size([2, 129])
```

**UNO**   We adopt the code from the open-source repository (Kossaifi et al., 2024) as the implementation for UNO. For the NS system, the weight composition of UNO is as follows:

```
lifting.fcs.0: torch.Size([256, 5])
lifting.fcs.1: torch.Size([64, 257])
fno_blocks.0.convs.0: torch.Size([64, 5185])
fno_blocks.0.channel_mlp.0: torch.Size([32, 65])
fno_blocks.0.channel_mlp.1: torch.Size([64, 34])
fno_blocks.1.convs.0: torch.Size([64, 5185])
fno_blocks.1.channel_mlp.0: torch.Size([32, 65])
fno_blocks.1.channel_mlp.1: torch.Size([64, 34])
fno_blocks.2.convs.0: torch.Size([128, 10369])
fno_blocks.2.channel_mlp.0: torch.Size([64, 129])
fno_blocks.2.channel_mlp.1: torch.Size([128, 66])
horizontal_skips.0.conv.weight: torch.Size([64, 64])
projection.fcs.0: torch.Size([256, 129])
projection.fcs.1: torch.Size([2, 257])
```

**WNO**   We adopt the code from the open-source repository (Tripura & Chakraborty, 2023) as the implementation for WNO. For the NS system, the weight composition of WNO is as follows:

```
conv.0.weights_a1 | Shape: torch.Size([40, 40, 5, 5])
conv.0.weights_a2 | Shape: torch.Size([40, 40, 5, 5])
conv.0.weights_h1 | Shape: torch.Size([40, 40, 5, 5])
conv.0.weights_h2 | Shape: torch.Size([40, 40, 5, 5])
conv.0.weights_v1 | Shape: torch.Size([40, 40, 5, 5])
conv.0.weights_v2 | Shape: torch.Size([40, 40, 5, 5])
conv.0.weights_d1 | Shape: torch.Size([40, 40, 5, 5])
conv.0.weights_d2 | Shape: torch.Size([40, 40, 5, 5])
conv.1.weights_a1 | Shape: torch.Size([40, 40, 5, 5])
conv.1.weights_a2 | Shape: torch.Size([40, 40, 5, 5])
conv.1.weights_h1 | Shape: torch.Size([40, 40, 5, 5])
conv.1.weights_h2 | Shape: torch.Size([40, 40, 5, 5])
conv.1.weights_v1 | Shape: torch.Size([40, 40, 5, 5])
conv.1.weights_v2 | Shape: torch.Size([40, 40, 5, 5])
conv.1.weights_d1 | Shape: torch.Size([40, 40, 5, 5])
conv.1.weights_d2 | Shape: torch.Size([40, 40, 5, 5])
conv.2.weights_a1 | Shape: torch.Size([40, 40, 5, 5])
conv.2.weights_a2 | Shape: torch.Size([40, 40, 5, 5])
conv.2.weights_h1 | Shape: torch.Size([40, 40, 5, 5])
conv.2.weights_h2 | Shape: torch.Size([40, 40, 5, 5])
conv.2.weights_v1 | Shape: torch.Size([40, 40, 5, 5])
conv.2.weights_v2 | Shape: torch.Size([40, 40, 5, 5])
conv.2.weights_d1 | Shape: torch.Size([40, 40, 5, 5])
conv.2.weights_d2 | Shape: torch.Size([40, 40, 5, 5])
conv.3.weights_a1 | Shape: torch.Size([40, 40, 5, 5])
conv.3.weights_a2 | Shape: torch.Size([40, 40, 5, 5])
conv.3.weights_h1 | Shape: torch.Size([40, 40, 5, 5])
conv.3.weights_h2 | Shape: torch.Size([40, 40, 5, 5])
conv.3.weights_v1 | Shape: torch.Size([40, 40, 5, 5])
conv.3.weights_v2 | Shape: torch.Size([40, 40, 5, 5])
conv.3.weights_d1 | Shape: torch.Size([40, 40, 5, 5])
conv.3.weights_d2 | Shape: torch.Size([40, 40, 5, 5])
w.0.weight | Shape: torch.Size([40, 40, 1, 1])
w.0.bias | Shape: torch.Size([40])
w.1.weight | Shape: torch.Size([40, 40, 1, 1])
w.1.bias | Shape: torch.Size([40])
w.2.weight | Shape: torch.Size([40, 40, 1, 1])
w.2.bias | Shape: torch.Size([40])
w.3.weight | Shape: torch.Size([40, 40, 1, 1])
w.3.bias | Shape: torch.Size([40])
fc0.weight | Shape: torch.Size([40, 5])
fc0.bias | Shape: torch.Size([40])
fc1.weight | Shape: torch.Size([128, 40])
fc1.bias | Shape: torch.Size([128])
fc2.weight | Shape: torch.Size([3, 128])
fc2.bias | Shape: torch.Size([3])
```

Since the parameters of normalization layers are determined by the dataset and are not controlled by the environment, we do not enable normalization layers in all operators (they are also disabled by default in the original code).

NEW

### I.1   COMPARISON WITH FINETUNING-FREE META-LEARNING APPROACHES

 Recent finetuning-free meta-learning approaches also achieve test-time adaptation without gradient-based optimization. For instance, methods like those proposed by (Gordon et al., 2018) and (Jiang et al., 2023). However, DynaDiff's paradigm differs from these works in several key aspects:

NEW

**Adaptation Mechanism (Modulation vs. Generation):** Prior works typically rely on a fixed, shared backbone network. Adaptation is achieved by generating or modulating a small subset of parameters, such as a final linear layer Gordon et al. (2018) or a context vector $c$ that conditions the backbone's dynamics Jiang et al. (2023). In contrast, DynaDiff generates the *complete* set of weights for an entire expert predictor from scratch. This allows the framework to adapt to more fundamental changes in dynamics, rather than only modulating the behavior of a fixed, shared model.

**Weight Representation (Vector vs. Structured Graph):** Our work introduces the **Weight Graph** to treat model weights as a structured data modality, preserving the inherent topological connectivity of the neural network architecture. This structured representation, combined with our proposed **Functional Loss**, differs from methods that generate unstructured weight vectors Gordon et al. (2018). As shown in our ablation studies, this structural-awareness is critical for effectively modeling the high-dimensional weight distribution.

**Target Domain (General vs. SciML-Specific):** DynaDiff is specifically tailored for cross-environment generalization in Scientific Machine Learning (SciML). Its **Dynamics-informed Prompter** is designed to extract physically meaningful features (e.g., energy, spectral patterns) from short observation sequences. This contrasts with the generic set-encoders used in few-shot classification or regression (Gordon et al., 2018; Jiang et al., 2023), enabling DynaDiff to capture informative features from complex physical dynamics even from limited observations ($L = 10$).

$$\text{Observation } (X_L) \xrightarrow{\text{Prompter}} \text{Prompt} \xrightarrow{\text{Diffusion Model}} \text{Latent Code } (z) \xrightarrow{\text{VAE Decoder}} \text{Expert Model Weights } (\theta)$$

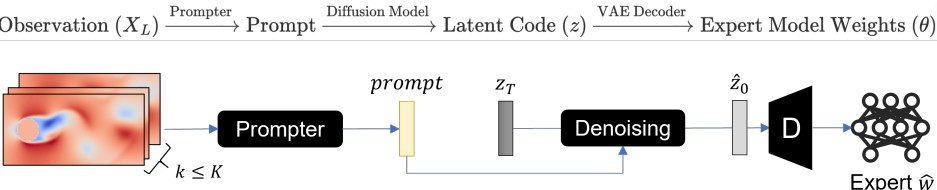

Figure 14: Simplest pipeline of DynaDiff.