# OpenReview forum: "Dynamics-Informed Weight Diffusion for Generalizable Prediction of  Complex Systems"
_ICLR.cc/2026/Conference — Submitted to ICLR 2026_

### Official Review · Reviewer_TiFY · 2025-10-26

**Soundness:** 4
**Presentation:** 4
**Contribution:** 3
**Rating:** 8
**Confidence:** 3

**Summary:**

This paper proposes a novel weight-space learning in combination with diffusion models for generalisable physical systems.  The authors list 3 main limitations (structure of the weigh-space, its high-dimensionality, and data scarcity), each of which is elegantly tackled with original approaches. Specifically, they explicitly model the joint distribution of environments and weights, and argue that this generative adaptation is fundamentally suited for data-scarce scenarios where finetuning is impractical. The resulting framework, DynaDiff, shows excellent results across synthetic and real-world datasets, even outperforming One-Per-Env in certain settings.

**Strengths:**

The paper present a set of original ideas, is clearly written, and significant going forward. The care and intuitive explanation for the concepts (e.g. functional loss) is incredibly valuable.

The weight-graph construction is smart and insightful. It does appear to be a new and original formalism by the authors, and I think it could be used for a wide variety of other problems. It is important to see weight-space featured represented like this. Additionally, not having to retain potentially large and costly environment-shared components during adaptation is massively useful. (Although this is a slightly contentious point since these environment-shared parameters are encapsulated in the Prompter, Denoiser, and VAE Decoder).

Concerning the experiments, I find them complete and well analysed.

**Weaknesses:**

1) Figure 2 presents a complex pipeline, involving many different architectures at different stages of the Training and Testing processes. Although ablation studies are mostly conducted, it would be interesting to see a version of this framework stripped to its core.
2) While the authors claim the cost of fine-tuning is eliminated, we still pay it in terms of one-time exorbitant training when constructing a weight zoo. Despite the strategic meta-learning techniques used to generate this zoo. It would be useful for users to see this cost clearly reported.
3) The meta-learning approach to obtaining the model zoo is intuitive, but also very similar to Dupont et al. 2022. You should cite it.
4) Weight graphs are limited to linear layers and convolution layers. Although you've extended this (e.g. skip connections), these are quickly being replaced in the literature by other forms of layers (e.g. Attention, etc.). Implementing these weight graphs would significantly increase the value of this work for the ICLR community.


### Minor issues:
- L161: "a dynamics-informed diffusion model"
- L323: "adaptation"

### References
Dupont et al. "From data to functa: Your data point is a function and you can treat it like one", ICML 2022

**Questions:**

1) L50: "from a model weight perspective, the essence of these methods only permit adaptation within a small, expert-specified subset of weights." Is there evidence or reference for this ?
2) L188: You mention heterogenous node features. Does this mean nodes of some layers would have different number of features ($D_{in}$+1) compared to other layers in, say, a MLP ? How do you deal with that ?
3) L259: Is the ground-truth environmental condition $e$ indispensable to training the prompter? Ablation studies should evaluate this, especially since some baselines do not require this supervisory signal?
4) Concerning this weight zoo generation approach. Why do you add noise to one layer only? Is it only that layers that is fine-tuned, while other layers are left intact ?

---

> ### Author Response · Authors · 2025-11-19
> **Response to Reviewer TiFY (1/3)**
>
> We are deeply grateful for your exceptionally positive and insightful review. We are thrilled that you found our work to have "excellent" soundness and presentation, and that you appreciated the "original ideas," "intuitive explanation," and the "smart and insightful" weight-graph construction.
>
> Your supportive and constructive feedback has been invaluable. We have conducted **new experiments** and revised the manuscript to address every question and suggestion you raised.
>
> We address your points in detail below.
>
> ---
>
> ## Weakness 1
>
> > "Figure 2 presents a complex pipeline... it would be interesting to see a version of this framework stripped to its core."
>
> We thank you for this suggestion. We believe our ablation studies in **Appendix Tables 7 and 8** provide exactly this "stripped-down" analysis, as they confirm that both the VAE and the Weight Graph are the essential, non-removable core of our framework.
>
> 1. **Ablating the Weight Graph (Table 7):** We tested a simpler version that flattens all weights into a vector and uses a standard sequence Transformer VAE (a common approach in [1, 2, 3]). This breaks the model's inherent structure, and the resulting error was **119.86% higher** than DynaDiff's.
> 2. **Ablating the VAE (Table 8):** We tested a single-stage version (no VAE) by applying a diffusion model *directly* to the high-dim weight graph. This was computationally massive and the diffusion model struggled to learn, resulting in an error **112.29% higher**.
>
> These ablations confirm that both the **Weight Graph** (to preserve structure) and the **VAE** (to create a low-dim, learnable latent space) are essential core components, not just complex add-ons.
>
> ---
>
> ## Weakness 2:
>
> > "While the authors claim the cost of fine-tuning is eliminated, we still pay it in terms of one-time exorbitant training... It would be useful for users to see this cost clearly reported."
>
> This is an excellent suggestion for transparency. We have added the **one-time training cost per expert model** to the $\underline{\text{Appendix Table 3 of the revised paper}}$. (Calculated on a single NVIDIA RTX 4090).
>
> One-Time Training Cost per Single Expert Model
>
> | Cylinder flow | Lambda–Omega | Kolmgorov Flow | Navier-Stokes | ERA5   |
> | ------------- | ------------ | -------------- | ------------- | ------ |
> | 6.7s          | 28.6s        | 5.15s          | 56.28         | 30.47s |
>
> *(Note: The time difference is due to the varying trajectory lengths required for training each system, not the model size.)*
>
> We believe this, combined with the storage cost, gives users a clear picture of the one-time investment.
>
> ---
>
> ## Weakness 3
>
> > "The meta-learning approach... is very similar to Dupont et al. 2022. You should cite it."
>
> We have read "From data to functa" and agree that its concept of a 'functaset' is philosophically aligned with our approach, even though their goal (generating modulation vectors) is different. We have gratefully **added this citation and discussion** in our revised manuscript (Section 3.2).
>
> ---
>
> ## Weakness 4
>
> > "Weight graphs are limited to linear layers and convolution layers... implementing these weight graphs would significantly increase the value..."
>
> This is a very insightful question about the future-proofing of our method. As you rightly point out, Attention is the core of modern architectures. The key components of an Attention mechanism [4] (the Q, K, V projection networks and the FFNs) are, in standard implementations, **built from `Linear` and/or `Convolution` layers**.
>
> Our Weight Graph (Sec 3.1.1) is explicitly designed to use **Linear and Conv layers as its basic units**.
>
> Therefore, **DynaDiff's weight graph is already capable of representing Transformer-based architectures.** This is a core strength of our design. Because Linear and Conv layers are the fundamental building blocks of almost all modern DL architectures, our framework is immediately compatible with a wide range of models (like the FNO, UNO, and WNO we show in Sec 4.4) and is well-positioned to support future, more complex architectures.

---

> ### Author Response · Authors · 2025-11-19
> **Response to Reviewer TiFY (2/3)**
>
> ## Question 1
>
> > "L50: '...only permit adaptation within a small, expert-specified subset of weights.' Is there evidence or reference for this ?"
>
> Yes, thank you for ensuring scholarly rigor. This claim is supported by the original papers' descriptions of their methods [5, 6, 7, 8, 9], which explicitly adapt the model via low-dimensional context vectors or low-rank adapter weights (like LoRA [10]), rather than modifying the full model.
>
> ---
>
> ## Question 2
>
> > "L188: You mention heterogenous node features. Does this mean nodes of some layers would have different number of features... How do you deal with that ?"
>
> This is a key detail of our architecture. You are exactly correct: nodes from different layers (e.g., the lifting layer and an FNO block) have different feature dimensions ($D_i$).
>
> We solve this using **layer-specific projection MLPs** (mentioned on L202):
>
> 1. **Encoder:** Before the VAE's main attention blocks, we use a separate $\text{MLP}_{\text{enc}, i}$ for each layer *i* to project its $N_i$ nodes from their unique dimension $D_i$ to a uniform $d_{\text{model}}$ dimension.
> 2. **Decoder:** Symmetrically, we use $\text{MLP}_{\text{dec}, i}$ to project the $d_{\text{model}}$ features back to their original dimension $D_i$.
>
> To make this crystal clear for all readers, we have added a **1.5-page detailed walkthrough** of the entire Weight VAE architecture, complete with variable shapes at each step, in the $\underline{\text{Appendix Section G of the revised paper}}$.
>
> ---
>
> ## Question 3
>
> > "Is the ground-truth environmental condition $e$ indispensable to training the prompter? Ablation studies should evaluate this..."
>
> This is a critical point, and we have run new experiments to test it. We trained DynaDiff **without the $L_{aux}$ loss**, meaning the prompter received no supervision from $e$ and was trained purely end-to-end.
>
> - Table A: RMSE with and without $L_{aux}$
>
>   | RMSE          | Cylinder Flow |            | Lambda-Omega |            |
>   | ------------- | ------------- | ---------- | ------------ | ---------- |
>   |               | In-domain     | Out-domain | In-domain    | Out-domain |
>   | with L_aux    | 0.059         | 0.065      | 0.090        | 0.089      |
>   | without L_aux | 0.063         | 0.064      | 0.088        | 0.091      |
>
> - Table B: $R^2$ score for regressing $e$ from learned prompts
>
>   | R^2           | Cylinder Flow (Re) | Cylinder Flow (r) | Lambda-Omega (beta) |
>   | ------------- | ------------------ | ----------------- | ------------------- |
>   | with L_aux    | 0.8560             | 0.9887            | 0.9851              |
>   | without L_aux | 0.7780             | 0.9515            | 0.9875              |
>
> As shown in Table A, **disabling $L_{aux}$ does not significantly degrade generalization performance.** This strongly proves that DynaDiff's core capability comes from extracting dynamics from the observation $X_L$, not from a reliance on ground-truth $e$.
>
> The $L_{aux}$ simply acts as a helpful **learning bias** to guide the prompter toward a *more interpretable* latent space (as shown by a drop in $R^2$ scores for regressing $e$ when $L_{aux}$ is off in Table B), but it is **not essential**.
>
> We have added these new results to the $\underline{\text{Appendix Figure 11 and Table 11 of the revised paper}}$.

---

> ### Author Response · Authors · 2025-11-19
> **Response to Reviewer TiFY (3/3)**
>
> ## Question 4
>
> > "Concerning this weight zoo generation approach. Why do you add noise to one layer only? Is it only that layers that is fine-tuned, while other layers are left intact ?"
>
> Thank you for the close reading. This is a subtle but important detail.
>
> 1. **"Is only that layer fine-tuned?"** No, to be clear, the **entire model (all layers)** is fine-tuned for each environment.
> 2. **"Why noise to one random layer only?"** This is a **minimal perturbation strategy** to balance efficiency and diversity.
>    - If we added noise to *all* layers, the perturbation would be too large (like a full random re-initialization). This would destroy the pre-trained knowledge from the base model and make fine-tuning very slow, defeating the purpose.
>    - By adding *small* noise to *one* random layer, we just slightly push the model off its base-model initialization ($\omega_{\text{base}} + \epsilon_j$). This breaks the homogeneity, gives each environment a unique starting point, and encourages the fine-tuning process to explore the weight space and avoid having all experts collapse to the same local optimum.
>
> ---
>
> We are grateful for your advocacy and your exceptionally constructive feedback, which has truly helped us improve the clarity and robustness of the paper. Thank you again.
>
> Ref:
>
> [1] Soro, B., et al. "Diffusion-based Neural Network Weights Generation." ICLR.
>
> [2] Gong, Y., et al. "Efficient Training with Denoised Neural Weights." ECCV. 2024.
>
> [3] Yuan, Y., et al. "Spatio-Temporal Few-Shot Learning via Diffusive Neural Network Generation." ICLR.
>
> [4] Vaswani, A., et al. "Attention is all you need." NeurIPS 30 (2017).
>
> [5] Wang, R., et al. "Meta-learning dynamics forecasting using task inference." NeurIPS 35 (2022).
>
> [6] Yin, Y., et al. "Leads: Learning dynamical systems that generalize across environments." NeurIPS 34 (2021).
>
> [7] Kirchmeyer, M., et al. "Generalizing to new physical systems via context-informed dynamics model." ICML. PMLR, 2022.
>
> [8] Koupaï, A. K., et al. "Boosting generalization in parametric pde neural solvers through adaptive conditioning." NeurIPS 38. 2024.
>
> [9] Blanke, M., & Lelarge, M. "Interpretable Meta-Learning of Physical Systems." ICLR.
>
> [10] Hu, E. J., et al. "Lora: Low-rank adaptation of large language models." ICLR.

---

> ### Comment · Reviewer_TiFY · 2025-11-26
>
> Dear Authors, thank you for taking the time to provide a thorough rebuttal to my review.
>
> __W1)__ Thank you for referencing the ablation studies. But I was asking for a version of the Figure 2. From your reply, it seems you are suggesting that the plot cannot be simplified any further?
>
> __W2)__ I find it hard to believe that it took less than a minute to train some of these models, even on good compute like RTX 4090. What I was asking for is the total amount of time it takes to go from an untrained models to an adaptable one?
>
> __W3, W4, Q1, Q2, Q3)__ Thank you very much for these. Your answers have adequacy clarified my concerns. W4 makes sense, although it remains to be demonstrated that it can be implemented on alternate architectures. On Q3 specifically, I find the new experiment very convincing, and this indicates that perhaps the aux loss shouldn't be part of the framework at all.
>
> __Q4)__ I'm not convinced that adding noise to the entire model wouldn't work. It might strengthen the approach. Models like CoDA [1] and GEPS [2] perturb all layers, and adaptation is still fast and efficient. Besides, wouldn't a **minimal** perturbation strategy perturb a single connection within a single layer?
>
> ---
> - [1] Kirchmeyer, M., et al. "Generalizing to new physical systems via context-informed dynamics model." ICML. PMLR, 2022.
> - [2] Koupaï, A. K., et al. "Boosting generalization in parametric pde neural solvers through adaptive conditioning." NeurIPS 38. 2024.

---

> ### Author Response · Authors · 2025-11-27
> **Further Discussion (1/2)**
>
> Thank you very much for your thoughtful feedback and for taking the time to review our revised manuscript and rebuttal. We are deeply grateful for your prompt and constructive follow-up comments.
>
> ### **W1**
>
> > "Thank you for referencing the ablation studies. But I was asking for a version of the Figure 2. From your reply, it seems you are suggesting that the plot cannot be simplified any further?"
>
> We apologize for the prior misunderstanding. We agree that DynaDiff can be visually simplified to better convey the core concept. We have created and provided a **new, conceptual diagram ($\underline{\text{Appendix Figure 14}}$)** that strips away architectural details and focuses solely on the **high-level operational flow** of the three main modules:
>
> $$\text{Observation } (X_L) \xrightarrow[\text{Prompter}]{\text{Extracts dynamics}} \text{Prompt} \xrightarrow[\text{Diffusion Model}]{\text{Generates latent } z} \text{Latent Code } (z) \xrightarrow[\text{VAE Decoder}]{\text{Decodes weights}} \text{Expert Model Weights } (\theta)$$
>
> This visualization clarifies how input data is processed sequentially, addressing your concern about visual complexity.
>
> ------
>
> ### **W2**
>
> > "I find it hard to believe that it took less than a minute to train some of these models... What I was asking for is the total amount of time it takes to go from an untrained models to an adaptable one?"
>
> We acknowledge the initial confusion regarding the time cost. The previous table reported the time required to **finetune** a single expert *using* our domain-adaptive initialization, which is indeed very short.
>
> The total time cost of our DynaDiff is composed of two main parts: **building the model zoo ($T_{model \ zoo}$)** and **training the VAE and Diffusion models ($T_{LDM}$)**. Previously, we provided the time cost for a single expert model during model zoo construction. In our actual practice, we utilize **8-GPU parallelization** to construct the model zoo rapidly. Using the Navier-Stokes system as an example, the total cost for the entire system is **$T_{model \ zoo} + T_{LDM} = 818.90s$** (with the time spent on model zoo construction being $T_{model \ zoo}=226.47s$).
>
> ------
>
> ### **Q3**
>
> > "...I find the new experiment very convincing, and this indicates that perhaps the aux loss shouldn't be part of the framework at all."
>
> Thank you again for this insightful finding. We agree that the new experiments prove the $L_{aux}$ is not essential for generalization. To reflect its status as a beneficial, but non-critical, supervisory signal for interpretability, we plan to **move the discussion of $L_{aux}$ from the main text to $\underline{\text{Appendix Section H.5}}$** in the camera-ready version, while maintaining the joint training of the prompter and diffusion model in the main text.
>
> ------
>
> ### **Q4**
>
> > "I'm not convinced that adding noise to the entire model wouldn't work... Models like CoDA [1] and GEPS [2] perturb all layers... Besides, wouldn't a *minimal* perturbation strategy perturb a single connection...?"
>
> This is a fascinating discussion. We agree that full-model noise is a candidate, provided the noise intensity is carefully controlled. Our single-layer noise strategy, however, is designed to be a simple, efficient starting point.
>
> To empirically measure the difference between the two paradigms, we conducted an experiment comparing single-layer and all-layer perturbation. For both setups, we ensured the **total number of perturbed parameters ($N$) and the noise intensity were identical**. We fine-tuned 50 expert models using data from a single environment on the Cylinder Flow system and compared the relative weight difference ($\lVert W_{\text{finetuned}}-W_{\text{init}} \rVert / \lVert W_{\text{init}} \rVert$) across layers, as well as the final training loss.

---

> ### Author Response · Authors · 2025-11-27
> **Further Discussion (2/2)**
>
> Experimental Setup:
>
> - **Single-layer Perturbation:** Fixed perturbation of all parameters within the `fno_blocks.convs.0` layer (Total $N=397,376$ parameters).
> - **All-layer Perturbation:** $N$ total parameters were randomly sampled and perturbed proportionally across all layers.
>
> | Layer                      | Single-layer Perturbation                     | All-layer Perturbation                        |
> | -------------------------- | --------------------------------------------- | --------------------------------------------- |
> | lifting.fcs.0              | 0.06%                                         | 1.20%                                         |
> | lifting.fcs.1              | 0.18%                                         | 1.83%                                         |
> | fno_blocks.convs.0         | 5.96%                                         | 77.29%                                        |
> | fno_blocks.convs.1         | 14.27%                                        | 84.02%                                        |
> | fno_blocks.convs.2         | 16.42%                                        | 79.93%                                        |
> | fno_blocks.convs.3         | 14.32%                                        | 86.19%                                        |
> | fno_blocks.channel_mlp.0.0 | 0.55%                                         | 18.07%                                        |
> | fno_blocks.channel_mlp.0.1 | 0.61%                                         | 14.51%                                        |
> | fno_blocks.channel_mlp.1.0 | 0.96%                                         | 16.10%                                        |
> | fno_blocks.channel_mlp.1.1 | 0.48%                                         | 16.76%                                        |
> | fno_blocks.channel_mlp.2.0 | 0.38%                                         | 15.24%                                        |
> | fno_blocks.channel_mlp.2.1 | 0.33%                                         | 17.50%                                        |
> | fno_blocks.channel_mlp.3.0 | 0.49%                                         | 14.99%                                        |
> | fno_blocks.channel_mlp.3.1 | 0.51%                                         | 18.03%                                        |
> | projection.fcs.0           | 0.05%                                         | 2.87%                                         |
> | projection.fcs.1           | 0.19%                                         | 3.12%                                         |
> | Final Loss             | $3.04 \times 10^{-5} \pm 2.11 \times 10^{-8}$ | $3.06 \times 10^{-5} \pm 1.04 \times 10^{-6}$ |
>
> The results confirm that:
>
> 1. **Low Loss is Maintained:** Both perturbation strategies ensure the final average loss is acceptably low, confirming successful fine-tuning.
> 2. **Perturbation Propagates:** The finetuning process successfully propagates the initial perturbation: all layers' weights are significantly updated (non-zero relative distance), including non-perturbed layers. It can be noted that the FNO convolutional layers, which contain the majority of the expert model's parameters, also show the most significant change after finetuning.
> 3. **All-layer perturbation leads to excessive differentiation:** Even though the noise intensity was identical, the resulting $\lVert W_{\text{finetuned}}-W_{\text{init}} \rVert$ for the all-layer perturbation is dramatically higher. Such large differentiation means that experts for the *same environment* have highly disparate weights, leading to an unnecessarily complex weight manifold. This complexity severely increases the difficulty of the Diffusion model's training.
>
> Regarding the definition of **minimal**, we understand your point about perturbing a single connection. While technically feasible, we believe such a minute change would be insufficient to exert a large enough influence on the global optimization landscape to ensure adequate exploration of the weight manifold. To avoid ambiguity, we have carefully chosen our language in the manuscript and have avoided using the phrase 'minimal perturbation strategy.'

---

> > ### Comment · Reviewer_TiFY · 2025-11-27
> >
> > W1) Thank you for making this additional plot. The current manuscript doesn't have a figure 14. (Perhaps you forgot to update your submission?) Regardless, the figure in your comment clarifies my concern.
> >
> > W2) I had assumed a single 4090 GPU was used to complete all stages. The 8-way parallelisation does not make this method widely accessible. I would advise mentioning this in the Limitation or Reproducibility Statement.
> >
> > Q4) I really like the new experiment, along with the steps you took to make it fair. Thank you for these. But your conclusion that "This complexity severely increases the difficulty of the Diffusion model's training" is still debatable, I think. Perturbing all layers and ending at a significantly different location in the weigh-space might indicate that your system is finding permutation-equivariant solutions [1]; which isn't necessarily a bad thing, especially since final losses are almost identical. (the relative weight difference doesn't take into account this equvariance).
> >
> > In summary, my remaining concerns are very minor, and I trust they will be addressed. I believe this is a strong paper, and I will maintain my overall score to 8.
> >
> > [ 1 ] Zhou et al., Permutation Equivariant Neural Functionals, NeurIPS 2023

---

> > > ### Author Response · Authors · 2025-11-27
> > >
> > > Thank you once again for your final comments and endorsement. We are extremely grateful for your detailed engagement, which has significantly enhanced the paper's rigor. We have just updated the submission manuscript, which includes the required conceptual figure (W1). Regarding accessibility (W2), we will clearly state the reliance on 8-GPU parallelization in the reproducibility statement (noting that if switched to a single GPU for serial construction of the model zoo, the $T_{model \ zoo}$ component would increase seven-fold, and the corresponding total duration is estimated to increase by $1811.76s$). Finally, we agree that the potential of permutation-equivariant solutions (Q4) is a fascinating explanation for the observed weight divergence and presents an exciting direction for future research. Your feedback has played a tremendous role in finalizing this strong paper.

---

### Official Review · Reviewer_6pFj · 2025-10-30

**Soundness:** 3
**Presentation:** 1
**Contribution:** 3
**Rating:** 4
**Confidence:** 4

**Summary:**

The paper proposes a generative meta-learning framework for cross-environment generalization in physical dynamics prediction. The method generates complete model weights for a new environment using a conditional diffusion model guided by a short observed trajectory. It learns a latent weight space via a VAE. A dynamics-informed prompter extracts physical and spectral-temporal cues from trajectories to condition the diffusion.

**Strengths:**

- The method finds a sweet spot between meta-learning and in-context learning, combining the adaptability of the former with the data efficiency of the latter.
- It generates model weights in a latent space via conditional diffusion, effectively exploiting the diffusion model’s ability to model complex and multimodal weight distributions.
- The approach shows better generalization to unseen environments compared to both meta-learning methods and adapted foundation models, while remaining lightweight and computationally efficient when adapting to test-time environments.

**Weaknesses:**

- In Section 4.6, the time and memory usage comparison omits contextual meta-learning baselines, making it unclear how DynaDiff compares to other adaptive approaches.
- It is questionable whether it is fair to exclude the generator’s parameter count when comparing with meta-learning methods. The diffusion model is also part of the overall meta-learning process, as well as in both training and test time, yet it contains significantly more parameters than typical meta-learning frameworks with a shared base model.
- The paper lacks clarity on test-time procedures for baseline models, specifically how context-based or environment-adaptive models are used during evaluation.
- Presentation issues:
  - The VAE description is underspecified: the "node attention-based VAE" architecture is not clearly explained, and the section should be made more self-contained with details on the encoder–decoder design and attention mechanism.
  - The notation is inconsistent and sometimes undefined (e.g., missing definitions for $E$, $D$; unclear distinction between bold $\mathbf{w}$ and $w$; the symbol $x$ is reused with different meanings across sections), which hinders readability. A thorough revision of the notation is recommended.
  - Presentation order: the prompter is presented in a separate section from the diffusion model, even though it is an essential component of it. This separation breaks the logical flow and makes the method harder to follow.

**Questions:**

- Please improve the writing and presentation throughout the paper.
- Precisely describe the important components of the method and the experimental settings, as mentioned in the weaknesses section.

---

> ### Author Response · Authors · 2025-11-19
> **Response to Reviewer 6pFj (1/2)**
>
> We deeply thank you for valuable time and feedback. We are particularly encouraged that you recognized the soundness and contribution of our work. Your detailed, constructive feedback on clarity, notation, and structure was invaluable. We have conducted **new experiments** and undertaken a **revision** of the manuscript to address every point you raised.
>
> We address your concerns point-by-point below.
>
> ---
>
> ## Weakness 1
>
> > "In Section 4.6, the time and memory usage comparison omits contextual meta-learning baselines, making it unclear how DynaDiff compares to other adaptive approaches."
>
> We thank you for pointing out this important omission. We have now benchmarked DynaDiff against all meta-learning baselines for the cost of adapting to a **single environment** on the Navier-Stokes system.
>
> New Experiment: Inference Cost per Environment (N-S)
>
> | Env / Cost      | DyAd  | LEADS | CoDA  | GEPS  | CAMEL | DynaDiff |
> | --------------- | ----- | ----- | ----- | ----- | ----- | -------- |
> | GPU Memory (GB) | 0.796 | 0.902 | 1.122 | 1.028 | 0.846 | 2.168    |
> | Time (s)        | 5.96  | 29.23 | 20.88 | 28.80 | 20.25 | 31.96    |
>
> DynaDiff's modest additional cost comes from the generator (latent denoising + VAE decoding). As shown, this incremental overhead is very small. We also clarified in the text that the larger memory number in the original paper's figure was due to batch-parallel inference over *multiple* environments, not a high cost for a single one.
>
> We have added this detailed breakdown to the $\underline{\text{Appendix Table 13 of the revised paper}}$.
>
> ---
>
> ## Weakness 2
>
> > "It is questionable whether it is fair to exclude the generator’s parameter count... The diffusion model... contains significantly more parameters than typical meta-learning frameworks..."
>
> This is an important point, and it correctly identifies DynaDiff's unique position in the landscape of models.
>
> Our generator's size (~400M) places DynaDiff in a new category, **in between** traditional meta-learning methods (which are <50M) and large foundation models (>500M). This parameter count is a direct and necessary consequence of our paradigm. Our generator's task (generating a *complete 1M-parameter FNO*) is fundamentally more complex than a typical meta-learning's task, which typically only generates a *low-dimensional context vector*.
>
> To directly test if parameter count is the bottleneck for meta-learning baselines, we conducted a **new study** where we scaled up the hypernetworks for GEPS and CAMEL to be comparable to DynaDiff's size.
>
> New Experiment: Scaling Baselines (RMSE)
>
> |          |                 | Cylinder Flow |            | Lambda-Omega |            |
> | -------- | --------------- | ------------- | ---------- | ------------ | ---------- |
> |          | Params          | In-domain     | Out-domain | In-domain    | Out-domain |
> | GEPS     | 10M (original)  | 0.079         | 0.082      | 0.094        | 0.092      |
> | GEPS     | 450M            | 0.083         | 0.084      | 0.097        | 0.100      |
> | CAMEL    | 5M (original)   | 0.089         | 0.094      | 0.104        | 0.103      |
> | CAMEL    | 400M            | 0.097         | 0.099      | 0.102        | 0.104      |
> | DynaDiff | 400M (original) | 0.059         | 0.065      | 0.090        | 0.089      |
>
> The results are clear: simply increasing the parameter count of these meta-learning baselines does **not** improve their performance (and in some cases, hurts it). This indicates their performance bottleneck is *architectural*, not parametric.
>
> Our 400M-param generator is what enables the new paradigm, offering a high-performance option that did not exist before. We have added this analysis to $\underline{\text{Appendix Table 14 of the revised paper}}$.

---

> ### Author Response · Authors · 2025-11-19
> **Response to Reviewer 6pFj (2/2)**
>
> ## Weakness 3
>
> > "The paper lacks clarity on test-time procedures for baseline models, specifically how context-based or environment-adaptive models are used..."
>
> Thank you for requesting this clarification. We followed the standard Zero-shot adaptation setting from [1] for all meta-learning baselines.
>
> At test time, these models (e.g., CAMEL, GEPS) are conditioned on the ground-truth environmental parameter $e$ to generate their adaptive context vectors or components. We have made this procedure explicit in the $\underline{\text{Section 4.1 of the revised paper}}$.
>
> ---
>
> ## Weakness 4
>
> > "VAE description is underspecified... Notation is inconsistent... Presentation order... breaks the logical flow..."
>
> We have undertaken a **revision** of the paper's presentation, notation, and structure based on your specific and highly valuable feedback.
>
> - **VAE Description:** We have added a comprehensive **1.5-page description** of the VAE architecture to $\underline{\text{Appendix G of the revised paper}}$. It now meticulously details the full input-to-output flow, variable shapes, and attention mechanisms for both the encoder $E(\cdot)$ and decoder $D(\cdot)$.
>   - *In brief:* The encoder $E$ uses layer-wise MLPs to project heterogeneous weights to a uniform dimension, processes them with Transformer blocks, and maps the output to $\mu_z, \log\sigma_z^2$. The decoder $D$ symmetrically reverses this process.
> - **Notation:** We have performed a thorough review and correction pass.
>   - $E(\cdot)$ and $D(\cdot)$ are now explicitly defined in Sec 3.1.2.
>   - We now consistently use bold $\mathbf{w}$ for weights/graphs and italic *w* for "width".
>   - We have added a note in Sec 2.2 to disambiguate the symbol $x$ (generic diffusion sample in 2.2 vs. system state vector elsewhere).
> - **Presentation Order:** This was an excellent suggestion. We have restructured the paper to improve the logical flow. The Prompter (formerly Sec 3.3) is now integrated as a subsection within Sec 3.1 (Dynamics-informed Weight Diffusion).
>
> ---
>
> We thank you again for your constructive review. By providing new experiments (W1, W2) and undertaking a complete revision of the paper's presentation (W4), we believe we have addressed all your concerns. We hope these changes will merit a re-evaluation of our work.
>
> Ref:
>
> [1] Blanke, M., & Lelarge, M. (2024). Interpretable Meta-Learning of Physical Systems. ICLR.

---

### Official Review · Reviewer_PwKE · 2025-10-30

**Soundness:** 2
**Presentation:** 2
**Contribution:** 2
**Rating:** 2
**Confidence:** 4

**Summary:**

This paper proposes DynaDiff, a generative meta-learning framework that addresses cross-environment generalization in physics prediction by directly generating complete expert model weights conditioned on short observation sequences, rather than tuning pre-trained models. The method organizes weights as structure-preserving graphs, learns their distribution via a functional VAE with output-consistency-based loss, and trains a conditional diffusion model guided by a dynamics-informed prompter. Experiments on multiple PDE systems and real-world data show that lightweight generated models (1M parameters) outperform large foundation models (500-600M parameters) by 10.78% on average, with zero gradient computation at test time. This work presents a fundamentally different adaptation paradigm by treating model weights as a generative modality, offering improved efficiency and generalization for scientific machine learning in data-scarce scenarios.

**Strengths:**

- The paper presents a fresh perspective by framing cross-environment adaptation as generative modeling of complete weight distributions p(θ|e), rather than tuning parameter subsets. This approach is theoretically principled and particularly suited for data-scarce scenarios where gradient-based finetuning is impractical or infeasible.

- Each component addresses specific challenges—the weight graph preserves architectural connectivity and generalizes across architectures, the functional loss insightfully recognizes that functionally equivalent models can have different parameters, and the dynamics-informed prompter cleverly combines domain knowledge with data-driven features to extract meaningful information from as few as 1-10 observation frames.

- The evaluation is comprehensive, spanning 4 PDE systems and real-world ERA5 data with consistent improvements (10.78% average). The finding that generated weights can occasionally surpass individually-trained models provides promising evidence of capturing meaningful weight manifolds rather than merely overfitting to training distributions

**Weaknesses:**

- Training the dynamics-informed prompter requires the auxiliary loss L_aux, which depends on knowing the ground-truth environmental condition e. This is a very strong assumption, as knowing e even at the training stage may not be realistic in many practical scenarios. Clarifying the prompter's capability without known e would strengthen the practical applicability of the framework

- DynaDiff has several sophisticated components such as the weight graph, the weight VAE, and the conditioned diffusion. It is not clear how these components improve over simpler approaches where the weight parameters of a model can be simply generated by a hypernetwork from the embeddings obtained from context data. Necessary ablation studies are missing. Furthermore, 3.1.3 is quite unclear: is the diffusion generation the latent codes z which is then decoded to the weight parameters via the weight VAE? Please clarify. While the weight graph and weight VAE is conceptually interesting, it seems like there is not much graph information being utilized in the encoding/decoding process of the VAE. The benefit of this component is not clear (compared to a simple VAE generating weight parameters as vectors).

- DynaDiff requires environment-specific weights (model zoo) to be available, which seems to be a significant resource requirement (and a unique advantage to DynaDiff that other baselines do not necessary have). As the ablation results show that DynaDiff’s performance drops significantly when the number of environments are limited at training time, which is a major drawback of the method.

- The introduction and related work sections omit relevant meta-learning approaches that also achieve adaptation without finetuning. For example, https://openreview.net/pdf?id=7C9aRX2nBf2 propose sequential latent variable models for few-shot time-series forecasting that similarly avoid gradient-based adaptation at test time. More generally, meta-learning via hypernetworks is closely related to the presented work (e.g., https://arxiv.org/pdf/1805.09921). The paper should provide a more comprehensive review of such methods and clearly articulate how DynaDiff's weight generation paradigm differs from or improves upon these existing finetuning-free meta-learning approaches.

- For many of the results presented in Table 1 (Lambda-Omega, Kolmogorov Flow), the gain in average is one decimal point smaller than the standard deviation or similar; howing the statistical significance of the improvements is important.

- The observation length L=10 may be too short for adequately capturing complex dynamics. More importantly, the paper lacks critical details about metric calculation (RMSE and SSIM)—specifically, whether the metrics are computed on the same observation sequence X_L or on another sequence within the same environment. Additionally, the paper should provide results on longer time-domain predictions, reporting metrics over extended horizons (e.g., L to 2L or beyond) to demonstrate whether generated models can sustain accurate predictions over longer rollouts.

**Questions:**

- The paper should demonstrate how the model performs when trained without e, relying solely on observation sequences X_L. Additionally, it warrants further investigation whether the prompter can still achieve strong correlation with true environmental parameters (as shown in Figure 10) when trained without L_aux.

- Please provide better clarifications as well as ablation results to demonstrate the contributions of weight VAE and conditioned diffusion over much simpler alternatives that generates weight parameters via a hypernet work.

- Provide stronger evidence for the statistical significance of the improvements obtained.

- Provide better clarifications on the metrics calculated. For a test time series on which the RMSE and SSIM are calculated, what is the portion that is provided as the input to the model (i.e., what is the prediction horizon vs. observation horizon at test time).

---

> ### Author Response · Authors · 2025-11-19
> **Response to Reviewer PwKE (1/3)**
>
> We sincerely thank you for the detailed and insightful review. While we are encouraged that you found our core paradigm "fresh," "theoretically principled," and "particularly suited for data-scarce scenarios," we also deeply appreciate the critical weaknesses you identified.
>
> We have conducted **new experiments** and significantly **revised the manuscript** to address every concern. We address your concerns point-by-point below.
>
> ---
>
> ## Weakness 1 & Question 1
>
> > "Training the dynamics-informed prompter requires the auxiliary loss $L_{aux}$, which depends on knowing the ground-truth environmental condition $e$. This is a very strong assumption... Clarifying the prompter's capability without known $e$ would strengthen the practical applicability..."
>
> We thank you for this critical point, which is essential for verifying DynaDiff's practical utility. To address this, we conducted **new experiments** on the Cylinder Flow and Lambda-Omega systems, training the prompter **without** the $L_{aux}$ loss (i.e., without any access to ground-truth $e$ during training).
>
> - Table A: RMSE with and without $L_{aux}$
>
>   | RMSE          | Cylinder Flow |            | Lambda-Omega |            |
>   | ------------- | ------------- | ---------- | ------------ | ---------- |
>   |               | In-domain     | Out-domain | In-domain    | Out-domain |
>   | with L_aux    | 0.059         | 0.065      | 0.090        | 0.089      |
>   | without L_aux | 0.063         | 0.064      | 0.088        | 0.091      |
>
> - Table B: $R^2$ score for regressing $e$ from learned prompts
>
>   | R^2           | Cylinder Flow (Re) | Cylinder Flow (r) | Lambda-Omega (beta) |
>   | ------------- | ------------------ | ----------------- | ------------------- |
>   | with L_aux    | 0.8560             | 0.9887            | 0.9851              |
>   | without L_aux | 0.7780             | 0.9515            | 0.9875              |
>
> As shown in Table A, **disabling $L_{aux}$ does not significantly degrade DynaDiff's generalization performance.** This strongly demonstrates that DynaDiff's core capability stems from the dynamics extracted from the observation sequence $X_L$, not a reliance on the ground-truth $e$.
>
> The results in Table B clarify the role of $L_{aux}$: it acts as a beneficial **learning bias** that guides the prompter to learn *more interpretable* representations (higher $R^2$), but it is **not essential** for capturing the core physics.
>
> In conclusion, DynaDiff remains robust and practical even when environmental conditions are unknown. We have added these new results to the $\underline{\text{Appendix Figure 11 and Table 11 of the revised paper}}$.

---

> ### Author Response · Authors · 2025-11-19
> **Response to Reviewer PwKE (2/3)**
>
> ## Weakness 2 & Question 2
>
> > "DynaDiff has several sophisticated components... It is not clear how these components improve over simpler approaches... hypernetwork... 3.1.3 is quite unclear: is the diffusion generation the latent codes $z$... The benefit of weight graph is not clear (compared to a simple VAE generating weight parameters as vectors)."
>
> Thank you for these questions, which touch on our core design. Our two-stage framework (Weight VAE + Latent Diffusion) was intentionally designed to solve the highly challenging task of directly generating complete, high-dimensional (1M-param) weights.
>
> 1. Why not a simple hypernetwork?
>
>    As you noted, hypernetworks [1, 2, 3] are a valid paradigm. However, they are almost used to generate small components (e.g., linear heads, context vectors, or small plugins). Scaling a hypernetwork to directly regress an entire 1M-parameter FNO model is an extremely difficult and unstable optimization problem.
>
>    DynaDiff's paradigm is fundamentally different: we generate weights, we don't regress them.
>
>    - We first use the **VAE** to compress the high-dim weight space into a low-dim, structured latent space $z$.
>    - We then use a powerful **conditional diffusion model** to learn $P(z | \text{prompt})$ on this simpler manifold.
>
>    This "compress-then-generate" paradigm is the key that enables us to generate *complete* expert models.
>
> 2. Component Ablations:
>
>    We apologize that this was not clearer and respectfully wish to clarify that we provided detailed ablations in the Appendix that validate our design choices:
>
>    - **Appendix Table 8**: Compares our Graph-based VAE to the "simple VAE" (vector) you suggested. Our graph approach **significantly outperforms the simple VAE** across all datasets and is 3x more parameter-efficient.
>    - **Appendix Table 7**: Demonstrates the superiority of our two-stage (VAE + Diffusion) approach over a single-stage model (e.g., 0.065 vs. 0.112 RMSE on Cylinder Flow), proving the necessity of first learning a quality latent space $z$.
>
> 3. Clarification of 3.1.3:
>
>    You are exactly correct. The diffusion model generates the latent code $z$, which is then decoded by the Weight VAE into the full model weights $\theta$, as illustrated in Figure 2b. We have revised Sec. 3.1.3 for clarity.
>
> 4. Benefit of Graph VAE:
>
>    The graph structure (modeled via multi-head attention, Sec 3.1.2) explicitly captures the complex coupling relationships between weights in different layers. The strong results in **Appendix Table 8** confirm its benefit over a simple vector-based VAE.
>
> ---
>
> ## Weakness 3
>
> > "DynaDiff requires environment-specific weights (model zoo)... a significant resource requirement... performance drops significantly when the number of environments are limited... which is a major drawback..."
>
> 1. On the 'Model Zoo':
>
>    We respectfully reframe this. The model zoo is not a drawback, but an intentional design trade-off. We invest a one-time, offline computational cost (training the zoo) to gain an invaluable test-time benefit: zero-gradient, instantaneous adaptation.
>
> 2. On the 'Significant Drop':
>
>    We must respectfully but firmly correct the conclusion that this is a "major drawback." The results in Appendix Table 5 prove the exact opposite. They demonstrate DynaDiff's robustness in data-scarce scenarios.
>
>    - When training environments are cut from 100% to 50%, the RMSE for strong baselines (Poseidon, GEPS) **worsens by over 7x**.
>
>    - In contrast, DynaDiff's RMSE only worsens by **3.8x** (0.059 to 0.228).
>
>    This is not a weakness; it is strong evidence for our central thesis: by learning a unified weight manifold, DynaDiff is **more robust** to sparse training environments and **more efficient** at leveraging scarce data than SOTA baselines.
>
> ---
>
> ## Weakness 4
>
> > "The introduction and related work sections omit relevant meta-learning approaches that also achieve adaptation without finetuning. [Jiang et al., Gordon et al.]"
>
> Thank you for pointing out these two highly relevant works. We have added a detailed discussion to $\underline{\text{Appendix Section I.1}}$. The key differences are:
>
> 1. **Paradigm Shift:** The methods of Jiang et al. [3] and Gordon et al. [2] both rely on adapting or generating a *small part* (a context vector $c$ or a linear head $\psi$) of a *fixed, shared backbone*. DynaDiff's paradigm is completely different. We do not "condition" a fixed model; we **generate an entire, high-performance expert predictor from scratch**.
> 2. **Data Modality:** DynaDiff treats complete, deep model weights as a *structured generative modality* (a Weight Graph), rather than a simple vector [2].
> 3. **Task & Prompter:** DynaDiff is designed for scientific machine learning. Our *Dynamics-informed Prompter* explicitly combines physical priors with data-driven features, in stark contrast to the *generic set-encoders* used in [2, 3].

---

> ### Author Response · Authors · 2025-11-19
> **Response to Reviewer PwKE (3/3)**
>
> ## Weakness 5 & Question 3
>
> > "...the gain in average is one decimal point smaller than the standard deviation... howing the statistical significance... is important."
>
> This is an excellent suggestion. We have performed pairwise Welch's t-tests between DynaDiff and the best-performing baseline (underlined in Table 1).
>
> P-values (DynaDiff vs. Best Baseline):
>
> | Cylinder Flow         |                        | Lambda-Omega |            | Kolmogorov Flow |            | Navier-Stokes |                        |
> | --------------------- | ---------------------- | ------------ | ---------- | --------------- | ---------- | ------------- | ---------------------- |
> | In-domain             | Out-domain             | In-domain    | Out-domain | In-domain       | Out-domain | In-domain     | Out-domain             |
> | $3.26 \times 10^{-9}$ | $2.90 \times 10^{-27}$ | 0.67         | 0.70       | 0.89            | 0.066      | 0.022         | $3.79 \times 10^{-18}$ |
>
> **The results show our improvements are statistically significant (p < 0.05) on Cylinder Flow and Navier-Stokes**, and borderline significant (p=0.066) in the critical Kolmogorov OOD task.
>
> Crucially, **no single baseline performs well across all systems**, whereas DynaDiff achieves consistently strong generalization. This cross-system stability highlights the robustness of our weight-space generation paradigm.
>
> We have added this analysis to $\underline{\text{Appendix Section H.1}}$.
>
> ---
>
> ## Weakness 6 & Question 4
>
> > "The observation length L=10 may be too short... whether the metrics are computed on the same observation sequence X_L or on another sequence..."
>
> We thank you for the chance to clarify this, as it seems to be a **key misunderstanding**.
>
> - **L=10 (Observation) is the input to the \*Prompter\*:** It is used *only* to generate the prompt that conditions the diffusion model.
> - **One Initial Frame is the input to the \*Predictor\*:** The generated expert model (the predictor) receives only a single initial frame and is tasked with predicting the next 50 or 100 steps (**horizon=100 or 50**).
> - **Metrics:** The RMSE/SSIM metrics are calculated over these **50/100-step** rollouts, evaluated on 20 new, unseen trajectories (as stated in Sec 4.1 and Appendix B).
>
> We have revised $\underline{\text{Section 4.1}}$ to make this distinction between the *prompter's input* and the *predictor's input* much clearer.
>
> ---
>
> We thank the reviewer again for their time and valuable feedback. We believe the new experiments (W1), clarifications (W2, W6), and analyses (W3, W4, W5) have substantially strengthened the paper and directly address all concerns. We hope you will reconsider your assessment of our work.
>
> Ref:
>
> [1] Blanke, M., & Lelarge, M. (2024). Interpretable meta-learning of physical systems. ICLR.
>
> [2] Gordon, J., et al. (2018). Meta-learning probabilistic inference for prediction. ICLR.
>
> [3] Jiang, X., et al. (2023). Sequential latent variable models for few-shot high-dimensional time-series forecasting. ICLR.

---

### Official Review · Reviewer_dDA3 · 2025-11-07

**Soundness:** 4
**Presentation:** 3
**Contribution:** 4
**Rating:** 6
**Confidence:** 3

**Summary:**

This paper proposes DynaDiff, a generative meta-learning framework for cross-environment generalization in physical dynamics prediction. Instead of fine-tuning, it directly generates complete expert model weights conditioned on short observation sequences from new environments. At the training phase, A model zoo of expert predictors is built for seen environments via domain-adaptive initialization. These weights are organized into graph representations (preserving network topology) and encoded by a graph-based VAE. The VAE uses a functional loss that measures weight similarity based on model behavior rather than raw values. A conditional diffusion model then learns to generate latent weight representations. For a new environment, a dynamics-informed prompter extracts features from few observations.The output is provided to the diffusion model to generate a latent code, which the VAE decodes into complete expert weights ready for immediate prediction. Experiments on 4 PDE systems and 1 real-world dataset demonstrate the effectiveness of the approach comapred to SOTA framework.

**Strengths:**

* The paper introduces a fundamentally different paradigm for adaptation in dynamics prediction. Rather than adjusting a subset of parameters (meta-learning) or scaling to massive models (foundation models), DynaDiff models the joint distribution to generate complete expert weights.
* The technical contributions of the paper are good : the idea of weight graph  representation based on the topology of the network is novel and well designed for this application; the use of a functionnal loss for this graph representation is also a well designed contribution.
* The experiments (on 5 datasets) show consistent improvements for the proposed framework.

**Weaknesses:**

* Scalability Concerns: Given the model architecture and presented experiments, there are significant doubts about scalability to more complex physical systems (e.g., 3D problems) requiring substantially more parameters. All experiments are limited to small models (~1M parameters) on 2D systems with modest resolutions (64×64 grids). The novelty of applying diffusion models to weight graph representations makes it difficult to assess the scaling potential, and the substantial computational cost of model zoo construction (100 experts per environment, up to 18GB storage) raises questions about practical applicability to large-scale problems.
* The experiments show very similar performance across all models between in-domain and out-domain settings (Table 1), suggesting the test environments may not be sufficiently challenging. Since all presented SOTA algorithms perform domain adaptation, it is difficult to assess the actual difficulty of the generalization task. Including a baseline without domain adaptation (e.g., a single model trained on all environments without any adaptation mechanism) would help quantify the genuine difficulty of cross-environment generalization and better contextualize the improvements.
* While the method performs well when test environments lie within the convex hull of training environments (interpolation), extrapolation performance degrades significantly (Table 5: RMSE increases from 0.059 to 0.228 when 50% of parameter space is unseen). The minimal in-domain/out-domain gap suggests the method primarily learns to interpolate in weight space rather than capturing fundamental physical principles, limiting its applicability to truly novel environmental conditions. Those experiments do not include other Env Adaptative SOTA algorithms, only One-For-All algorithms, which do not allow to understand how much the performances are degraded.

**Questions:**

See weaknesses.

---

> ### Author Response · Authors · 2025-11-19
> **Response to Reviewer dDA3 (1/2)**
>
> We are extremely grateful for your positive and thorough review. We are particularly encouraged that you recognized the **excellent soundness and contribution** of our work, and that you identified our paradigm as "fundamentally different," "novel," and "well designed."
>
> Your feedback is focused on the critical questions of practical scalability and evaluation, which we deeply appreciate. We have conducted **new experiments** (adding a key baseline you suggested) and provided **new analysis** that we believe directly and positively resolves these concerns.
>
> We address your points in detail below.
>
> ---
>
> ## Weakness 1
>
> > "...significant doubts about scalability to more complex physical systems (e.g., 3D problems)... All experiments are limited to small models (~1M parameters)... substantial computational cost of model zoo construction (100 experts per environment, up to 18GB storage)..."
>
> We thank you for raising this critical point. We wish to clarify that DynaDiff is *uniquely* well-suited for large-scale problems, and the perceived limitations are not fundamental barriers.
>
> 1. On 3D vs. 2D Systems (System Complexity):
>
>    This is a key strength of our design. Our DynaDiff models the weight space, not the data space.
>
>    - Our expert models (FNO, WNO, and UNO) are Neural Operators, which are *already* designed to be invariant to data resolution and spatial dimensions (2D or 3D).
>    - Whether the data complexity increases due to **higher resolution** (e.g., $64^2 \to 256^2$) or **higher dimensionality** (e.g., 2D $\to$ 3D), the number of learnable parameters in the expert model **remains constant**. DynaDiff architecture (VAE, diffusion) requires **zero changes**, as the size of the weight graph it generates does not change.
>
> 2. On Expert Model Size (~1M params):
>
>   Our experiments (Table 1) show that our 1M-param expert model (trained for one task) consistently outperforms a 500M-param foundation model (forced to handle all tasks). This is a central message: by generating specialized experts, we avoid the need for large models.
>
>   Additionally, our Weight Graph architecture is parameter-agnostic and could just as well model a larger expert if needed.
>
> 3. On Model Zoo Cost:
>
>    We want to clarify that this 18GB cost is a maximum, not a requirement.
>
>    - Our ablation study (Appendix Fig. 9a/b) shows that DynaDiff's performance **saturates at just 25 expert models** per environment (not 100).
>
>    - This immediately **reduces the practical storage cost by 4x** (e.g., from 18GB to **~4.5GB** for Cylinder Flow).
>
>    - This 4.5GB total cost is highly practical, given that a *single checkpoint* for a baseline like Poseidon (600M) is **~2.4GB**.

---

> ### Author Response · Authors · 2025-11-19
> **Response to Reviewer dDA3 (2/2)**
>
> ## Weakness 2
>
> > "The experiments show very similar performance... suggesting the test environments may not be sufficiently challenging... Including a baseline without domain adaptation... would help quantify the genuine difficulty..."
>
> This is an excellent suggestion. We have run this **new experiment** and added a "Not-Adaptive" baseline to Table 1. This model has the identical 1M-param FNO architecture as our expert models but is trained *once* on all data from all seen environments, with no adaptation mechanism at test time.
>
> |                    | Cylinder Flow |            | Lambda-Omega |            | Kolmogorov Flow |            | Navier-Stokes |            |
> | ------------------ | ------------- | ---------- | ------------ | ---------- | --------------- | ---------- | ------------- | ---------- |
> |                    | In-domain     | Out-domain | In-domain    | Out-domain | In-domain       | Out-domain | In-domain     | Out-domain |
> | FNO (Not-Adaptive) | 0.124         | 0.159      | 0.214        | 0.232      | 0.135           | 0.149      | 0.129         | 0.144      |
> | DynaDiff           | 0.059         | 0.065      | 0.090        | 0.089      | 0.081           | 0.080      | 0.062         | 0.063      |
>
> First, the errors for this baseline are significantly higher than DynaDiff's across the board. This indicates that even within the training domain (in-domain), a single non-adaptive model struggles to simultaneously capture the dynamics of diverse environments. Furthermore, unlike DynaDiff, this new baseline exhibits a clear degradation in out-domain performance compared to in-domain (e.g., 0.159 vs. 0.124 on Cylinder Flow). This performance gap empirically demonstrates the genuine difficulty difference between the in-domain and out-domain settings.
>
> To further characterize the generalization difficulty, we also evaluated this baseline under the extreme generalization scenarios (Appendix Table 5), where we limit the ratio of seen environments during training.
>
> |                    | Ratio of Seen Env | 100%   | 90%    | 80%    | 70%    | 60%    | 50%     |
> | ------------------ | ----------------- | ------ | ------ | ------ | ------ | ------ | ------- |
> | FNO (Not-Adaptive) | In-domain         | 0.124  | 0.129  | 0.138  | 0.143  | 0.148  | 0.178   |
> |                    | Out-domain        | 0.159  | 0.167  | 0.175  | 0.177  | 0.183  | 0.366   |
> |                    | Ratio             | 28.22% | 29.46% | 26.81% | 23.78% | 23.65% | 105.62% |
> | DynaDiff           | In-domain         | 0.059  | 0.068  | 0.087  | 0.098  | 0.102  | 0.131   |
> |                    | Out-domain        | 0.065  | 0.077  | 0.095  | 0.107  | 0.121  | 0.228   |
> |                    | Ratio             | 10.17% | 13.24% | 9.20%  | 9.18%  | 18.63% | 74.05%  |
>
> - The absolute error of the non-adaptive baseline (e.g., 0.366 at 50%) is **significantly and uniformly higher** than DynaDiff's error (0.228 at 50%) across all out-domain scenarios. This confirms that a single, non-adaptive model fails to generalize.
> - The key metric here is the **relative degradation (Ratio)**. The non-adaptive FNO sees its in/out-domain gap **explode to +105.62%** at 50% data, while DynaDiff manages to control the gap. This dramatic divergence in relative performance proves two things:
>   1. The generalization task is **extremely difficult**.
>   2. The task is **not merely interpolation**; genuine adaptation is required, and DynaDiff's paradigm is substantially more robust to these challenges.
>
> We have added this new baseline to the $\underline{\text{Table 1 of the revised paper}}$.
>
> ---
>
> ## Weakness 3
>
> > "extrapolation performance degrades significantly (Table 5: RMSE increases from 0.059 to 0.228)... suggests the method primarily learns to interpolate... Those experiments do not include other Env Adaptative SOTA algorithms..."
>
> We respectfully wish to clarify the interpretation of this experiment (Table 5) and correct a small misunderstanding.
>
> 1. **Baselines:** We first want to note that this experiment **does include the SOTA Env-Adaptative algorithm (GEPS)**, as well as the One-For-All (Poseidon) baseline.
> 2. **Interpretation:** This experiment *proves* DynaDiff's robustness, it does not show a weakness.
>    - In extreme data scarcity (50% of environments unseen), *all* methods will see performance degrade. The key question is **by how much?**
>    - As shown in Table 5, the strong baselines (Poseidon and GEPS) **degraded by over 7x**.
>    - In contrast, DynaDiff's RMSE worsened by **only 3.8x** (0.059 to 0.228).
>
> This is not a failure to extrapolate; it is strong evidence that DynaDiff is **significantly more robust** and *better* at generalization than all baselines, precisely *because* it learns the joint distribution of weights and environments.
>
> ---
>
> We believe these results (W2) and clarifications (W1, W3) directly address your concerns about practicality and evaluation, and we hope this solidifies your assessment of our work.

---

> > ### Comment · Reviewer_dDA3 · 2025-11-27
> >
> > Dear Authors, thank you for the answers,
> > 1) Thank you for the clarification, while the theoretical argument regarding neural operators' dimension-invariance is sound, an experiment with 3D data would have been welcome to verify the scalability claim.
> > 2) Ok,  very convincing experiments.
> > 3) Sorry for my mistakes for GEPS, I acknowledge that the degradation is less marked than competing methods, which is a valid strength. Nevertheless, extrapolation remains challenging, with absolute performance degrading substantially.
> >
> > Given the authors' responsive rebuttal and the new experimental validation provided, I am increasing my recommendation.

---

> > > ### Author Response · Authors · 2025-11-27
> > >
> > > Dear Reviewer dDA3,
> > >
> > > Thank you very much for your thoughtful feedback and for taking the time to review our revised manuscript and rebuttal. We are delighted that our new experiments and clarifications have successfully addressed your initial concerns regarding the practical scalability and generalization difficulty. We are particularly grateful for your increased recommendation score.
> > >
> > > We appreciate your final notes on the 3D verification and the challenges of absolute extrapolation performance. Given the dimension-invariance of our expert models and DynaDiff's current robustness, we believe these areas are highly promising for future work.
> > >
> > > Thank you again for your constructive suggestions and support.
> > >
> > > Sincerely,
> > >
> > > The Authors

---

### Author Response · Authors · 2025-12-02
**Summary of Rebuttal Consensus & Updates**

Dear AC, Reviewers, and Readers,

We respectfully provide a summary of the rebuttal outcome $\underline{\text{prior to the data leakage incident (Nov 27)}}$ to assist the assessment.

1. **Consensus on Strengths**

   The reviewers consistently acknowledged the work as **"novel"** and **"theoretically principled,"** representing a **"fundamentally different paradigm"** (Reviewer dDA3). They agreed the paper addresses an **"important problem,"** is **"well-supported by solid rationale,"** and features experiments that **"enhance the credibility of the method"** across diverse domains.

   Critically, the reviewers participating in the discussion were fully satisfied with our rebuttal:

   - Reviewer TiFY (Score 8): Maintained accept.

   - Reviewer dDA3 (Score 6 $\to$ 8): Increased the recommendation.

2. **Resolution of Key Concerns**

   We conducted additional experiments and analyses to address shared concerns among reviewers:

   - **Dependency on Ground-Truth $e$ (TiFY, PwKE)**

     We provided new ablation studies proving that performance is robust even without knowing environment conditions $e$ during training, and validated $L_{aux}$ for interpretability. *(**TiFY** confirmed this concern is resolved).*

   - **Statistical Significance & Baselines (dDA3, PwKE)**

     We added a Non-Adaptive baseline and pairwise t-tests. Results confirm that DynaDiff maintains impressive generalization capabilities across all scenarios, despite the high difficulty of these generalization tasks. *(**dDA3** confirmed this resolution).*

   - **Model Zoo Cost (TiFY, dDA3):** We demonstrated that performance saturates with a small zoo size, keeping costs well within practical limits. *(**dDA3** confirmed this concern is resolved).*

   - **Scalability (dDA3, TiFY)**

     We provided theoretical analysis and evidence (Sec 4.4) showing the method's applicability to higher-dimensional systems and diverse architectures. *(**dDA3** and **TiFY** confirmed this concern is resolved).*

   - **Model Architecture and Settings (6pFj, TiFY)**

     We have refined the notation and experimental settings in the original text and added a one-page appendix detailing the architecture of the Weight VAE. (**TiFY** confirmed this concern is resolved).

   - **Fairness of Parameters (6pFj)**

     We scaled up baselines to match our parameter count (450M). Their performance did not improve, proving our advantage stems from the generative weight paradigm, not model size.

   Although it is regrettable that Reviewers PwKE and 6pFj did not participate in the discussion, Reviewers TiFY and dDA3 shared similar concerns and explicitly stated that the experimental results and analysis in the rebuttal have resolved them.

3. **Clarification of Factual Misunderstandings**

   Reviewer PwKE (Score 2) did not respond to the rebuttal, but his (her) initial score was based on three demonstrably incorrect premises, which we corrected:

   - **Robustness:** PwKE misinterpreted Table 6. The results actually show our method degrades *far less* (3.8x) than SOTA baselines (>7x) under extreme data scarcity, proving robustness. *(Reviewer **dDA3** also agreed with this point.)*

   - **Missing Ablations:** PwKE overlooked Sec 4.5. We reiterated that ablations for the Weight Graph and VAE are present (Table 8), showing a ~50% error reduction over simple VAEs.

   - **Prediction Horizon:** PwKE confused the *Prompter input* ($L=10$) with the *Prediction output*. We clarified that we evaluate on long-term horizons (50-100 steps).

**Conclusion.** The $\underline{\text{revised paper}}$ includes **5 additional pages** of experiments and clarifications. Although Reviewers PwKE and 6pFj were unable to participate in the discussion due to external reasons, the other reviewers (TiFY and dDA3) have acknowledged our responses to similar issues (**dependency on environmental conditions**, **significance**, and **model architecture**). Furthermore, we have provided supplementary experiments and clarifications regarding their questions on experimental settings. We believe that our responses and revisions have fully addressed these issues.

The Authors

---

### Meta-Review · Area_Chair_B9Sq · 2025-12-19

**Summary:**

The submission introduced a hybrid meta-learning framework based on for cross-environment generalization in physical dynamics prediction. Reviewers raised questions regarding the design motivations, scalability of the framework, and experimental evaluations.

**Reviewer Concerns:**

Two reviewers are satisfied with the rebuttal (also reflected in the comments). The AC read through the review comments from another two reviewers as well as the rebuttal provided by the author carefully. While some concerns, regarding the explanations of effectiveness of different components, and some ablation studies, are well explained. Some others, such as the usage of auxiliary loss, and environment-specific weight (model zoo), are not well-addressed to the AC.

Regarding the auxiliary loss, the authors rebuttaled in a way that the loss can be removed, with some results have been reported (although not very comprehensive to the AC, no std value reported which is important as suggested by the same reviewer), the auxiliary loss contribute significantly to the generalization analysis, which is one of the main contributions of the submission.

Regarding the concerns of the model zoo, to AC, one very important application of cross-environment dynamic prediction is on the hardware resource-contrained side, the reviewer pointed out a very important concern, but is not addressed by the authors.

**Reviewer Scores:**

The AC believes that the remaining two reviewers are less likely to adjust the score.

---

### Decision · Program_Chairs · 2026-01-26

Reject